# KGML-ag: A Modeling Framework of Knowledge-Guided Machine Learning to Simulate Agroecosystems: A Case Study of Estimating N₂O Emission using Data from Mesocosm Experiments

Licheng Liu[1], Shaoming Xu[2], Jinyun Tang[3], Kaiyu Guan[4,5,6], Timothy J. Griffis[7], Matthew D. Erickson[7], Alexander L. Frie[7], Xiaowei Jia[8], Taegon Kim[1, 9], Lee T. Miller[7], Bin Peng[4,5,6], Shaowei Wu[10], Yufeng Yang[1], Wang Zhou[4,5], Vipin Kumar[2], Zhenong Jin[1]*

[1]Department of Bioproducts and Biosystems Engineering, University of Minnesota, Saint Paul, MN, 55108, USA
[2]Department of Computer Science and Engineering, University of Minnesota, Minneapolis, MN, 55455, USA
[3]Climate and Ecosystem Sciences Division, Lawrence Berkeley National Laboratory, Berkeley, CA 94720, USA
[4]Agroecosystem Sustainability Center, Institute for Sustainability, Energy, and Environment, University of Illinois at Urbana-Champaign, Urbana, IL 61801, USA
[5]Department of Natural Resources and Environmental Sciences, University of Illinois at Urbana-Champaign, Urbana, IL 61801, USA
[6]National Center for Supercomputing Applications, University of Illinois at Urbana-Champaign, Urbana, IL 61801, USA
[7]Department of Soil, Water, and Climate, University of Minnesota, Saint Paul, MN 55108, USA
[8]Department of Computer Science, University of Pittsburgh, Pittsburgh, PA, 15260, USA
[9]Department of Smart Farm, Jeonbuk National University, Jeonju, Jeollabuk-do, 54896, Republic of Korea
[10]School of Physics and Astronomy, University of Minnesota, Minneapolis, MN, 55455, USA

*Correspondence to*: Zhenong Jin (jinzn@umn.edu)

**Abstract.**

Agricultural nitrous oxide (N₂O) emission accounts for a non-trivial fraction of global greenhouse gases (GHGs) budget. To date, estimating N₂O fluxes from cropland remains a challenging task because the related microbial processes (e.g., nitrification and denitrification) are controlled by complex interactions among climate, soil, plant and human activities. Existing approaches such as process-based (PB) models have well-known limitations due to insufficient representations of the processes or uncertainties of model parameters, and to leverage recent advances in machine learning (ML) a new method is needed to unlock the "black box" to overcome its limitations such as low interpretability, out-of-sample failure and massive data demand. In this study, we developed a first-of-kind knowledge-guided machine learning model for agroecosystems (KGML-ag), by incorporating biogeophysical/chemical domain knowledge from an advanced PB model, *ecosys*, and tested it by comparing simulating daily N₂O fluxes with real observed data from mesocosm experiments. The Gated Recurrent Unit (GRU) was used as the basis to build the model structure. To optimize the model performance, we have investigated a range of ideas, including: 1) Using initial values of intermediate variables (IMVs) instead of time series as model input to reduce data demand; 2) Building hierarchical structures to explicitly estimate IMVs for further N₂O prediction; 3) Using multitask learning to balance the simultaneous training on multiple variables; and 4) Pretraining with millions of synthetic data generated from *ecosys* and fine tuning with mesocosm observations. Six other pure ML models were developed using the same mesocosm data to serve as the benchmark for the KGML-ag model. Results show that KGML-ag did an excellent job in reproducing the mesocosm

N$_2$O fluxes (overall r$^2$ = 0.81, and RMSE = 3.6 mg N m$^{-2}$ day$^{-1}$ from cross-validation). Importantly KGML-ag always outperforms the PB model and ML models in predicting N$_2$O fluxes, especially for complex temporal dynamics and emission peaks. Besides, KGML-ag goes beyond the pure ML models by providing more interpretable predictions as well as pinpointing desired new knowledge and data to further empower the current KGML-ag. We believe the KGML-ag development in this study will stimulate a new body of research on interpretable ML for biogeochemistry and other related geoscience processes.

## 1 Introduction

Nitrous oxide (N$_2$O), with its global warming potential $273 \pm 118$ times greater than that of carbon dioxide (CO$_2$) for a 100-year time horizon, is one of the major greenhouse gases (IPCC6; Forster et al., 2021). The increasing rate of atmospheric N$_2$O concentration during the period 2010-2015 is 44% higher than during 2000-2005, mainly driven by increased anthropogenic sources that have increased total global N$_2$O emissions to ~17 Tg N yr$^{-1}$ (Syakila and Kroeze, 2011; Thompson et al., 2019). It is estimated that approximately 60% of the contemporary N$_2$O emission increases are from agriculture management at global scale (Pachauri et al., 2014; Robertson et al., 2014; Tian et al., 2020), but the estimation uncertainty can exceed 300% (Barton et al., 2015; Solazzo et al., 2021). Quantifying N$_2$O emissions from agricultural soils is extremely challenging, partly because the related microbial processes, mainly about incomplete denitrification and nitrification, are controlled by many environment and management factors such as temperature/water conditions, soil/crop properties, and N fertilization rate, all of which together have collectively led to large temporal and spatial variabilities of N$_2$O emissions (Butterbach-Bahl et al., 2013; Grant et al., 2016).

Process-based (PB) models are often used for simulating N$_2$O fluxes from agroecosystems, but they have some inherent limitations, including incomplete knowledge of the processes, low accuracy due to the under-constrained parameters, expensive computing cost, and rigid structure for further improvements, that we could not resolve by using PB model itself. For example, an advanced agroecosystem model, *ecosys* (Grant et al., 2003, 2006, 2016), simulates N$_2$O production rates through nitrification and denitrification processes when oxygen (O$_2$) is limited, with equations considering the influence from related substrate concentrations (e.g., NO$_2^-$, N$_2$O, and CO$_2$), nitrifier and denitrifier populations, and soil thermal, hydrological physical and chemical conditions. The produced N$_2$O accumulates, transfers in gaseous phase, aqueous phase, over different soil layers, and eventually exchanges with atmosphere at the soil surface. Other PB models, including DNDC (Zhang et al., 2002; Zhang and Niu, 2016), DAYCENT (Del Grosso et al., 2000; Necpálová et al., 2015), and APSIM (Keating et al., 2003; Holzworth et al., 2014), have also included processes to simulate N$_2$O production, but adopt different parameterizations using static partition parameters to estimate N$_2$O emission from nitrification, and other empirical parameters to control the influence on nitrification from soil water content, pH, temperature and substrate concentrations. Besides, N$_2$O is intimately connected with the soil organic carbon (SOC) dynamics, because soil nitrifiers and denitrifiers interact strongly with aerobic and anaerobic heterotrophs that process SOC evolution, and all of these microbes are driven by shared environmental variables

including soil temperature, moisture, redox status, and physical and chemical properties (Thornley et al., 2007). As expected, these connections make it difficult for PB models, even the most advanced ones like *ecosys*, to find sufficient representations of the physical and biogeochemical processes or obtain enough data to calibrate a large number of model parameters with strong spatio-temporal variations. Thus, novel approaches are needed for addressing the big challenge of agricultural $N_2O$ flux simulations.

Machine learning (ML) models can automatically learn patterns and relationships from data. Recent studies have investigated the potential to predict agricultural $N_2O$ emission with ML models, including random forest (RF, Saha et al., 2021), metamodelling with extreme gradient boosting (XGBoost) (Kim et al., 2021), and deep learning neural network (DNN) (Hamrani et al., 2020). Notably, Hamrani et al. (2020) compared nine widely used ML models for predicting agricultural $N_2O$. That study pointed out that the long short term memory (LSTM) model with recurrent networks containing memory cells as building blocks will be most suitable for $N_2O$ predictions, but the challenge remains with respect to the ability of capturing the sharp peak of $N_2O$ fluxes and lag time between N fertilizer application and the emission peak. Although there is an increasing interest in leveraging recent advances in machine learning, capturing this opportunity requires going beyond the ML limitations, including limited generalizability to out-of-sample scenarios, demand for massive training data, and low interpretability due to the "black-box" use of ML (Karpatne et al., 2017). PB models with their transparent structures built by representations of physical and biogeochemical processes, seem to be exact complementary to ML models. Thus, combining the power of ML model and PB model understanding innovatively is likely a path forward.

The above need to integrate ML and PB models can be potentially addressed by the newly proposed framework of Knowledge-guided Machine Learning (KGML) models. In the review by Willard et al. (2021), five research frontiers have been identified regarding the development of KGML for diverse disciplines including earth system science, they are: 1) Loss function design according to physical or chemical laws (Jia et al., 2019, 2021; Read et al., 2019); 2) Knowledge-guided initialization through pretraining ML models with synthetic data generated from PB models (Jia et al., 2019, 2021; Read et al., 2019); 3) Architecture design according to causal relations or adding dense layers containing domain knowledge (Khandelwal et al., 2020; Beucler et al., 2019, 2021); 4) Residual modeling with ML models to reduce the bias between PB model outputs and observations (Hanson et al., 2020); and 5) Other hybrid modeling approaches combining PB and ML models (Kraft et al., 2021). These recent advances in KGML pave the pathway to a more efficient, accurate and interpretable solution for estimating $N_2O$ fluxes from the agroecosystem.

In this study, we present a first-of-its-kind attempt of developing a KGML for agricultural GHG fluxes prediction (KGML-ag) with knowledge-guided initialization and architecture design, and demonstrate the potential of KGML-ag with a case study on quantifying $N_2O$ flux observed by a multi-year mesocosm experiments. We designed the KGML-ag structure based on the causal relations of related $N_2O$ processes informed by an advanced agroecosystem model, *ecosys* (Grant et al., 2003, 2006,

2016). We used the synthetic data generated from *ecosys* to design the KGML-ag input/output, and to pre-train the KGML-ag model to learn the basic patterns of each variable. Observations from multi-season controlled-environment mesocosm chambers (Miller, 2021, thesis; Miller et al., 2021, in review) were used to refine the pretrained KGML-ag and evaluate the model performance. Since there is limited literature that guides the development of KGML-ag and not a one that directly addressed GHG fluxes, we investigated a range of ideas to optimize the model performance, including: 1) Using initial values of intermediate variables (IMVs) instead of sequences as model input to reduce data demand; 2) Building hierarchical structures to explicitly estimate IMVs for further $N_2O$ prediction; 3) Using multitask learning to balance the simultaneous training on multiple variables; and 4) Pretraining with millions of synthetic data generated from *ecosys* and fine tuning with mesocosm observations. Although we evaluated the KGML-ag models with real measurements only from a mesocosm experiment, the lessons learned from the development process and various KGML-ag structures can be transferred to other data, other variables and large scale simulations, therefore have broader implications on further KGML related research in agriculture. We believe this study will stimulate a new body of research on interpretable machine learning for biogeochemistry and other related topics in geoscience.

## 2 Methods

### 2.1 Experimental design overview

To develop and evaluate the KGML-ag models and compare their performance with pure ML models, we designed the following experiments:

1) With the synthetic data, we developed and pretrained multiple KGML-ag models to learn general patterns and interactions among variables, and evaluated their model performance (Fig. S2, Table 1);

2) With the observed data, we finetuned multiple KGML-ag models to adapt real-world situations, and evaluated their model performance (Fig. 2-3; Fig. S3-5; Table 2-3);

3) We further benchmarked KGML-ag models and uncertainties with other pure ML models without considering temporal dependence, including Decision Tree (DT), Random Forest (RF), Gradient Boosting (GB) from the sklearn package (https://scikit-learn.org/stable/), Extreme Gradient Boosting (XGB) from the XGBoost package (https://xgboost.readthedocs.io/en/latest/) and a 6-linear-layer artificial neural network (ANN) with the mesocosm experiment data by 10 times ensemble experiments (Fig. 4-5; Fig. S6-8);

4) We conducted a few small experiments to further investigate how various model configurations, such as the pretraining process, data augmentation and IMV initial values would influence KGML-ag model performance (Table 3).

 **2.2 KGML-ag structure development**

 **2.2.1 Generating synthetic data with *ecosys***

We generated synthetic data using a PB model, *ecosys*. The *ecosys* model is an advanced agroecosystem model constructed
from detailed biophysical and biogeochemical rules instead of using empirical relations (Grant et al., 2001). It represents $N_2O$
evolution in the microbe-engaged processes of nitrification-denitrification using substrate kinetics that are sensitive to soil
nitrogen availability, soil temperature, soil moisture, and soil oxygen status (Grant and Pattey 2008). Two groups of microbial
populations, autotrophic nitrifiers and heterotrophic denitrifiers, produce $N_2O$ with specific competitive or cooperative
relations in *ecosys* when $O_2$ availability fails to meet $O_2$ demand for their respiration, and $NO_2^-$ become alternative electron
acceptors. $N_2O$ transfer within soil layers and from soil to the atmosphere is driven by concentration gradient using diffusion-
convection-dispersion equations, in the forms of gaseous and aqueous $N_2O$ under control of volatilization-dissolution (Grant
et al., 2016). Unlike the pipeline model described by Davidson et al. (2000) , which mainly considers the correlations of $N_2O$
production with nitrogen availability and of $N_2O$ emissions with soil water content, *ecosys* enables integrative effects of energy,
water, nitrogen availability on $N_2O$ production and $N_2O$ transfer via the microbial population dynamics and their interactions
with soil, plant, and atmospheric dynamics, under diverse meteorological and anthropogenic disturbances (e.g. runoff,
drainage, tillage, irrigation, soil erosion). Many previous studies have demonstrated its robustness in simulating agricultural
carbon and nitrogen cyclings at different spatial/temporal scales, and under different management practices (Grant et al., 2003,
2006, 2016; Metivier et al., 2009; Zhou et al., 2021).  For the agricultural ecosystems in the US Midwest, whose simulations
are used for synthetic data in this study, the performance of *ecosys* on $CO_2$ have been extensively benchmarked, including $CO_2$
exchange (daily Reco $R^2$ = 0.80-0.86; daily NEE, $R^2$ = 0.75-0.89) and leaf area index (LAI, $R^2$ = 0.78) from six flux towers,
USDA census reported corn yield ($R^2$ = 0.83) and soybean yield ($R^2$ = 0.80), satellite-derived GPP for corn ($R^2$ = 0.83) and
soybean ($R^2$ = 0.85) in the US Midwest (Zhou et al., 2021). In addition, *ecosys* model can capture the dynamics and magnitude
of $N_2O$ flux in hourly frequency ($R^2$ = 0.2-0.4 and RMSE = 0.1-0.2 mg N $m^{-2}\,h^{-1}$ in Grant et al., 2008; $R^2$ = 0.28-0.37 and
RMSE = 0.2-0.28 mg N $m^{-2}\,h^{-1}$ in Grant et al., 2003), and in various ecosystems (e.g. agriculture soil in Grant et al., 2006,
2008; forest in Grant et al., 2010; and grassland in Grant et al., 2016). Therefore, *ecosys* is an appropriate choice of
domain knowledge provider and synthetic data generator in the development of KGML models. We generated daily synthetic
data including $N_2O$ flux and 76 IMVs (e.g. $CO_2$ flux from soil, layerwise soil $NO_3^-$ concentration, layerwise soil temperature,
and layerwise soil moisture; detailed in Table S1) from *ecosys* simulations for 2000-2018 over 99 randomly selected counties
in Iowa, Illinois, and Indiana, USA. We used hourly meteorological inputs (downward shortwave radiation, air temperature,
precipitation, relative humidity, and wind speed) from the phase 2 of North American Land Data Assimilation System
(NLDAS-2, Xia et al., 2012) and layerwise soil properties (e.g. bulk density, texture, pH, SOC concentration) from the
SSURGO database (Soil Survey Staff, 2020) as inputs to *ecosys*. Crop management except N fertilization rates were configured
to the same settings as mesocosm experiments (described in Sec 2.2.2). To increase the variability in synthetic data, we
implemented 20 different N fertilization rates ranging from 0 to 33.6 g N m$^{-2}$ (i.e. 0 to 300 lb N ac$^{-1}$) in each simulation of 99
counties, and more detailed information for model setup refers to Zhou et al. (2021).

The generated synthetic data were then processed for further use by KGML-ag development. Meanwhile, the hourly weather
forcings were converted to seven daily variables, including the maximum air temperature (TMAX_AIR, $^{\circ}$C), difference
between the maximum and the minimum air temperature (TDIF_AIR, $^{\circ}$C), the maximum humidity (HMAX_AIR, fraction),
difference between the maximum and the minimum humidity (HDIF_AIR, fraction), surface downward shortwave radiation
(RADN, W m$^{-2}$), precipitation (PREC, mm day$^{-1}$), and wind speed (WIND, m s$^{-1}$). Six soil properties were retrieved from the
SSURGO database, including total averaged (depth weighted averaged for all layers) bulk density (TBKDS, Mg m$^{-3}$), sand
content (TCSAND, g kg$^{-1}$), silt content (TCSILT, g kg$^{-1}$), pH (TPH), cation exchange capacity (TCEC, cmol$^{+}$ kg$^{-1}$) and soil
organic carbon (TSOC, g C kg$^{-1}$); and two crop properties were retrieved, including planting day of the year (PDOY) and crop
type (CROPT, 1 for corn and 0 for soybean). Finally, each synthetic data sample has daily N$_2$O flux, 76 selected IMVs, 7
weather forcings (W), 1 N fertilization rate (FN, g N m$^{-2}$) and 8 soil/crop properties (SCP) (Fig. 1.a; Table S1). The periods
from April 1st to July 31st (122 days) were selected to cover the mesocosm observations (around 30 days before and 90 days
after N fertilizer date). The total amount of synthetic data sample is 122 days x 18 years x 99 counties x 20 N fertilizer rates
(about 4.3 million data points). We randomly selected the samples from 70 counties for training, 10 counties for validation,
and 19 counties for testing.
**2.2.2 Mesocosm experiments for KGML-ag model fine-tuning and evaluation**
Observations were acquired from a controlled-environment mesocosm facility on the St. Paul campus of the University of
Minnesota. Soil samples were sourced in 2015 from a farm in Goodhue County, MN (44.2339$^{\circ}$ N and 92.8976$^{\circ}$ W), which had
been under corn-soybean rotation for 25 years. Six chambers with a soil surface area of 2 m$^2$ and column depth of 1.1 m were
used to plant continuous corn during 2015-2018 and monitor the N$_2$O flux response to different precipitation treatments. The
experiment also measured other environmental variables including air temperature and photosynthetically active radiation
(PAR), which were controlled to mimic the outdoor ambient environment. Granular urea fertilizer was hand broadcasted and
incorporated to a depth of 0.05 m to each chamber at a rate of 22.4 g N m$^{-2}$ (200 lb N ac$^{-1}$) on May 1st of 2015, May 4th of
2016 and May 3rd of 2017, and 10.3 g N m$^{-2}$ (92 lb N ac$^{-1}$) on May 8th of 2018. Corn hybrid (DKC-53-56RIB) were hand
planted to a depth of 0.05 m in two rows spaced 0.76 m apart 3-5 days after fertilizer application, at a seeding rate of 35,000
seeds ac$^{-1}$ in 2015 to 2017, and 70,000 seeds ac$^{-1}$ in 2018 but thinned upon emergence to ensure 100 percent emergence at
35,000 seeds ac$^{-1}$. Crops were harvested at the end of September by cutting the stover five inches above the soil. Hourly N$_2$O
fluxes (mg N m$^{-2}$ h$^{-1}$) and CO$_2$ fluxes (g C m$^{-2}$ h$^{-1}$) were measured using non-steady-state flux chambers with a CO$_2$ analyzer
(LI-10820 for 2016 and LI-7000 for 2017 and 2018, LI-COR Biosciences, Lincoln, NE) and a N$_2$O analyzer (Teledyne
M320EU, Teledyne Technologies International Corp, Thousand Oaks, CA) (Detail method can be retrieved from Fassbinder
et al., 2012, 2013). We also collected soil moisture at 15 cm depth (VWC as abbreviation of volumetric water content, m$^3$ m$^-$
[3]), weekly 0-15 cm depth soil $NO_3^- + NO_2^-$ concentration ($NO_3^-$ for short in the following text, g N $Mg^{-1}$), soil $NH_4^+$
concentration ($NH_4^+$, g N $Mg^{-1}$), and related environment variables including air temperature, radiation, humidity and soil/crop
properties from three growing seasons during 2016-2018 and six mesocosm chambers (Fig. S1). The magnitude of $N_2O$ flux
and $NO_3^-$ soil concentration and their responses following fertilizer application from this mesocosm experiment are slightly
higher than several field studies of agricultural soils (Fassbinder et al., 2013; Grant et al., 1999, 2006, 2008, 2016; Hamrani et
al., 2020; Venterea et al., 2011). More details about the mesocosm facility and experimental design can be found in the thesis
of Miller L. (2021).

The observed data were then processed to fine-tune and evaluate the KGML-ag models. The $N_2O$ flux and four IMVs and
weather variables were collected from the measurements in the selected period (i.e., April 1st to July 31st). Weekly $NO_3^-$ (short
for soil $NO_3^-$ within 0-15 cm depth), and $NH_4^+$ (short for soil $NH_4^+$ within 0-15 cm) were linearly interpolated to the daily time
scale on days containing VWC (short for soil VWC in 15 cm) data. Hourly air temperature, net radiation, $N_2O$ (short for $N_2O$
fluxes from soil), $CO_2$ (short for $CO_2$ fluxes from soil) and VWC were resampled to daily scale. All SCP were derived from
mesocosm measurements except that TCEC was derived from the SSURGO database according to the soil origin. We used the
leave-one-out cross-validation (LOOCV) method for the evaluation process. Each time we used five chambers' data for model
finetuning and another one chamber data for validation. For example, if we used chamber 1-5 to train the model, then chamber
6 would serve as the out-of-sample data to validate the results.  Only the validation results would be presented in our study.

To reduce overfitting and increase the generalization of the trained model based on the small amount of mesocosm data, we
applied the following method to augment the experimental measurements and weather forcings to 1000 times larger by
sampling hourly data and averaging them to daily scale. In this method, 16 hours (or maximum valid hours) of data are
randomly selected from 24 hours of data to compute their mean as the daily value. Since up to 2/3 of the day is covered by the
selected data (16 hours /24 hours), the augmented daily values should be representative enough for the source day with slight
variations from each other. Furthermore, the observation ratio, (24 hours - missing hours) / 24 hours, can be used as the weights
in loss function to inject the data quality information in model optimization. If the day has more than 16 hours missing values,
we consider the observations in that day as not trustworthy and drop the day by setting the weight to 0. This method can not
only augment the data to 1000 times larger but also deal with the missing values in observed data inherently. The total amount
of observed mesocosm data and related weather forcings are augmented to 122 days x 3 years x 6 chambers x 1000 data
samples in this study.

**2.2.3 Gated Recurrent Unit (GRU) as the basis of KGML-ag**
Hamrani et al. (2020) compared different models and reported that LSTM provided the highest accuracy in predicting $N_2O$
fluxes, because $N_2O$ flux is time dependent by its production/consumption nature and LSTM simulates target variables by

considering both current and historical states. The LSTM model, proposed by Hochreiter and Schmidhuber (1997), uses a cell state as an internal memory to preserve the historical information. At each time step, it creates a set of gating variables to filter the input and historical information and then uses the processed data to update the cell state. Similar to LSTM, GRU is a gated recurrent neural network but only keeps one hidden state (Cho et al., 2014). Though simpler than LSTM, GRU is proved to have similar performance (Chung et al., 2014). Our preliminary test on synthetic data for $N_2O$ prediction showed that GRU indeed provided similar or higher accuracy and model efficiency under different model settings than LSTM (Table S2). This is possible because simpler models with fewer weights and hyperparameters are more robust in combating the overfitting problem. Therefore, we choose GRU as the basis of KGML-ag development.

**2.2.4 Incorporating domain knowledge to the development of KGML-ag**

To quantitatively reveal the correlations between $N_2O$ fluxes and IMVs and guide the KGML-ag development, we conducted feature importance analysis by a customized 4-layer GRU ML model (Fig. 1b). Each layer of the model has a GRU cell with 64 hidden units. The 4-layer structure makes the model deeper and capable of capturing complex interactions. Between each GRU cell, 20% of the output hidden states are randomly dropped by replacing them with zero values (so called 20% dropout) to avoid overfitting. A linear dense layer is used to map the final output to $N_2O$. We first trained GRU models using synthetic data with different combinations of IMVs as inputs to predict the $N_2O$ fluxes (original-test, Table S2). The feature importance analysis of well-trained models was then implemented by replacing one input feature with a Gaussian noise with mean $\mu=0$ and standard deviation $\sigma=0.01$, while keeping others untouched (new-test). The importance score was calculated by the new-test's root mean square error (RMSE) (replacing one feature) minus the original-test's RMSE (no replacing). RMSE was calculated by $\sqrt{\frac{\Sigma_1^N(y_i-y_i')^2}{N}}$ where $N$ is the total number of observations across time and space, $y_i$ is i-th measurement from synthetic data or observed data and $y_i'$ is its corresponding prediction.

To find important variables for $N_2O$ flux prediction in an ideal situation where all variables are available, we conducted a feature importance analysis for GRU models with all IMVs and basic inputs including FN, 7 W and 8 SCP (Fig. S2a). Results indicated that flux variables including $NH_3$, $H_2$, $N_2$, $O_2$, $CH_4$, evapotranspiration (ET) and $CO_2$ had significant influence on the model performance. Variables ranked high in feature importance analysis are considered with priority during model development. To develop a functionable KGML-ag, we further investigated the feature importance of four IMVs that are available from mesocosm observations including $CO_2$, $NO_3^-$, VWC and $NH_4^+$, which were ranked 7th, 20th, 58th, 60th respectively in 92 input features of synthetic data (Fig. S2a). We used these four available IMVs to create two input combinations: 1) $CO_2$ flux, $NO_3^-$, VWC and $NH_4^+$ (IMVcb1), and 2) $NO_3^-$, VWC and $NH_4^+$ (IMVcb2). The objective of building IMVcb2 was to investigate the importance of the highly ranked variable $CO_2$ flux (by removing it from the inputs), and the impact of mixing-up flux and non-flux variables on model performance. We tested the feature importance of the GRU models built with IMVcb1 and IMVcb2 to check whether they would help in $N_2O$ prediction (Fig. S2b-c). All the feature

importance results above indicated the correlation intensity between $N_2O$ and many other variables, which would help the
KGML-ag model development and interpretation in this study (rest of this section and Sec. 3.1), and would guide future $N_2O$
related measurements and KGML model development (discussed in Sec. 4.3).

Next we used the knowledge learned from synthetic data to develop the structure of KGML-ag (Fig. 1c-d). Previous studies
for KGML models have used physical laws, e.g., conservation of mass or energy, to design the loss function for constraining
the ML model to produce physically consistent results (Read et al., 2019; Khandelwal et al., 2020). However, for complex
systems like agroecosystems, it is challenging to incorporate physical laws, such as mass balance for $N_2O$, into the loss function
due to the incomplete understanding of the processes and the lack of mass balance related data for validation. An alternative
solution is to incorporate such information in the design of the neural network (Willard et al., 2021). Effectiveness of such an
approach was demonstrated by Khandelwal et al. (2020) in the context of modeling stream flow in a river basin using Soil &
Water Assessment Tool (SWAT). They used a hierarchical neural network to explicitly model IMVs (e.g., soil moisture, snow
cover) and their relationships with the target variable (streamflow) and showed that this model is much more effective than a
neural network that attempts to directly learn the relationship between input drivers and the target variables. Following this
idea, we identified four desired features of an effective KGML-ag model, including: 1) We used initial values instead of
sequence of the IMVs from synthetic data or observed data to provide a solid starting state for the ML system and reduce the
IMV data demand, and then used the rest of the data to further constrain the prediction of IMVs; 2) We built a hierarchical
structure based on the structure of process representation in *ecosys* to first predict IMVs and then simulate $N_2O$ with predicted
IMVs; 3) We trained all variables together using multitask learning to reach the best prediction scores, which generalized the
model and incorporated interactions between IMVs and $N_2O$; 4) We initialized the KGML-ag model by pretraining with
synthetic data before using real observed data to transfer physical knowledge, which further reduced the demand on large
training samples and aided in faster convergence for fine-tuning.

To meet these desired features, we proposed two KGML-ag models (Fig. 1c-d). The first model, KGML-ag1, is a hierarchical
structure containing two modules to simulate IMVs and $N_2O$ sequentially. Each module is a 2-layer 64 units GRU ML model.
The inputs to the module of the KGML-ag1 model for IMV predictions (KGML-ag1-IMV module) are FN, 7W and 8SCP
together with the initial values of IMVs, and the outputs are IMV predictions. The inputs to the module of the KGML-ag1
model for $N_2O$ predictions (KGML-ag1-$N_2O$ module) are FN, 7W, 8SCP and predicted IMVs from KGML-ag1-IMV, and the
output is the target variable $N_2O$. Linear dense layers were coded for both modules to map output states to IMVs or $N_2O$. The
dropout method was applied to drop 20% of the state output between GRU cells and dense layers. The second model, KGML-
ag2, is also a hierarchical structure similar to KGML-ag1, but has multiple KGML-ag2-IMV modules to explicitly simulate
IMVs by tuning them separately in the fine-tuning process (discussed in Sec. 2.2.5). Each KGML-ag2-IMV module in KGML-
ag2 is a 2-layer 64 units GRU cell with the inputs of FN+7W+8SCP and one IMV initial value, and the output of one IMV
prediction. The KGML-ag2-N$_2$O module collects the IMV predictions from KGML-ag2-IMV modules and predicts the N$_2$O
with inputs of FN+7W+8SCP and predicted IMVs.

**2.2.5 Strategies for pretraining and fine-tuning processes**

To increase the efficiency of the training process, we used the Z-normalization ($\frac{(X-\mu)}{\sigma}$, where $X$ is the vector of a particular
variable over all the data samples in the data set; $\mu$ is the mean value of $X$; $\sigma$ is the standard deviation of $X$) method to normalize
each variable separately on synthetic data. Then the scaling factors ($\mu$, $\sigma$) derived from *ecosys* synthetic data for each variable
were used to Z-normalize observed data into the same ranges as synthetic data. As mentioned in Sec. 2.2.1, the TDIF_AIR,
HDIF_AIR were used instead of absolute min temperature (TMIN_AIR) and humidity (HMIN_AIR). This is done because
TMIN_AIR and HMIN_AIR follow similar trends as TMAX_AIR and HMAX_AIR, making Z-normalization numerically
poorly defined. Using the difference between maximum and minimum can provide a clearer information of daily air
temperature/humidity variation.
During the pretraining process, we initialized the IMV of KGML-ag using the first day value of synthetic IMV time series.
Adam optimizer with a start learning rate of 0.0001 was used for the training process. The learning rate would decay by 0.5
times after every 600 training epochs. At each epoch, synthetic data samples were randomly shuffled before being input to the
model to predict N$_2$O (and IMVs if any). The mean square error (MSE) loss (calculation was equal to the square of RMSE) or
sum of MSE loss (if multitask learning) between predictions and *ecosys* synthetic observations were calculated to optimize the
weights of GRU cells. After the training process updated the model's weights, the validation process was performed to evaluate
the model performance based on untouched samples with RMSE and the square of Pearson correlation coefficient ($r^2$). $r^2$ was
calculated as $\frac{(\Sigma_i\ (y_i{'}-\underline{y_i{'}})(y_i-\underline{y_i}))^2}{\Sigma_i\ (y_i{'}-\underline{y_i{'}})^2(y_i-\underline{y_i})^2}$, where $y_i$ is the i-th measurement from synthetic data or observed data, $y_i{'}$ is its
corresponding prediction, $\underline{y_i}$ is the mean of the measurement $y$ in diagnosing space and $\underline{y_i{'}}$ is the mean of the predicted $y'$ in
diagnosing space. If both validated $r^2$ and RMSE were better than the best values in previous epochs, the updated model in this
epoch would be saved. Normalized RMSE (NRMSE, calculated by RMSE/(max-min) of each variable observation) was
introduced to evaluate IMV predictions between variables with different value ranges.
During the fine-tuning process, we used estimated IMV initial values of 1.0 g C m$^{-2}$, 0.2 m$^3$ m$^{-3}$, 0.0 g N Mg$^{-1}$, and 20.0 g N
Mg$^{-1}$ for CO$_2$, VWC, NH$_4^+$, and NO$_3^-$ respectively, from starting day (April 1st) to the day before the first day of real
observations, as input to KGML-ag models. Then the first-day values of observed IMVs were input into KGML-ag during the
rest days of the period as IMV initial values. In addition, as described in Sec. 2.2.2, we used a data augmentation method to
augment the total amount of data 1000 times larger for the fine-tuning process. The purpose of this data augmentation method
was to increase the generalization of the fine-tuned model and to overcome the overfitting due to small sample size. The mask

matrix was elementarily multiplied to the output matrix to calculate the MSE, $r^2$ and RMSE only for days with observations. The similar optimizer was used with an initial learning rate of 0.00005 and decay fraction of 0.5 per 200 epochs. Other training/validation methods in each epoch were similar to the pretraining process. Specifically, in the KGML-ag1 model finetuning process, we first froze the KGML-ag1-N₂O module and only trained the KGML-ag1-IMV module for IMVs. After finishing the KGML-ag1-IMV module training, we froze the KGML-ag1-IMV module and trained the KGML-ag1-N₂O module for N₂O. In the KGML-ag2 fine-tuning process, the similar freezing method was used but different KGML-ag2-IMV modules were trained separately one by one.

**2.3 Development environment description**

We used the Pytorch 1.6.0 (https://pytorch.org/get-started/previous-versions/) and python 3.7.9 (https://www.python.org/downloads/release/python-379/) as the programing environment for the model development. In order to use the GPU to speed-up the training process, we installed cudatoolkit 10.2.89 (https://developer.nvidia.com/cuda-toolkit). A desktop with Nvidia 2080 super GPU was used for code development and testing. The Mangi cluster (https://www.msi.umn.edu/mangi) from High Performance Computing of Minnesota Supercomputing Institute (HPC-MSI, https://www.msi.umn.edu/content/hpc) with 2-way Nvidia Tesla V100 GPU was used in training processes which consumed longer time and bigger memories.

**3 Results**

**3.1 Pretraining experiments using synthetic data from *ecosys***

In the pretraining stage, the GRU model with 76 IMVs achieved the best performance in predicting N₂O fluxes ($r^2$=0.98, RMSE =0.54 mg N m$^{-2}$ day$^{-1}$ and normalized RMSE (NRMSE) = 0.01) on the test set of synthetic data generated from *ecosys* (Table 1). The high performance was due to some flux IMVs such as NH₃, H₂, O₂, CO₂ and ET, which are highly correlated to N₂O (Fig. S2a), were used as input to the model. The good performance of GRU with all IMVs indicates that ML models are able to perfectly mimic *ecosys* when sufficient information about IMVs is available. The GRU model with only basic input of N fertilizer rate, 7 weather forcings, and 8 soil/crop properties (FN+7W+8SCP) had the accuracy of $r^2$=0.89 and RMSE = 1.37 mg N m$^{-2}$ day$^{-1}$ (Table 1). The relatively low performance is likely because this model failed to capture several highly nonlinear pathways that are employed by ecosys to predict N₂O (e.g., one influence pathway from precipitation to N₂O can be: Precipitation → soil moisture → N components solubility/concentration → nitrification/denitrification rate/amount → soil N₂O concentration → gas N₂O flux). When adding sequences of IMV combinations (i.e., IMVcb1 of CO₂ flux, NO₃⁻, NH₄⁺ and VWC, and IMVcb2 of NO₃⁻, NH₄⁺ and VWC), the GRU models performed slightly better than the GRU model using only basic inputs, achieving $r^2$ of 0.92 and 0.90, respectively (Table 1). The KGML-ag1 with IMVcb1 and IMVcb2 initial values provided better performance (both $r^2$ = 0.90) than GRU with basic input and comparable performance to the GRU with inputs

of IMVcb1 and IMVcb2 sequence. Besides, KGML-ag1 provided predicted IMVs of $CO_2$, $NO_3^-$, $NH_4^+$, and VWC with $r^2$ over
0.91, and NRMSE below 0.06 (Table 1). KGML-ag2 also provided comparable $N_2O$ performance but relatively better IMVs
performance of $r^2$ over 0.92 and NRMSE below 0.05. Results indicated that KGML-ag models with IMV initial values as extra
input performed similar or better than pure ML models in synthetic data.

**3.2 KGML-ag evaluation using observed data from mesocosm**

After being fine-tuned with observed data, KGML-ag1 had $N_2O$ prediction overall accuracy of $r^2$=0.81 and RMSE=3.6 mg N
$m^{-2}$ day$^{-1}$, while non-pretrained GRU model provided $r^2$=0.78 and RMSE=4.0 mg N $m^{-2}$ day$^{-1}$, and pretrained GRU model
provided $r^2$=0.80 and RMSE=3.77 mg N $m^{-2}$ day$^{-1}$ (Table 3). The time series of $N_2O$ predictions from KGML-ag1 and the non-
pretrained GRU model were further compared (Fig. 2), from which we found at least two advantages of using KGML-ag1 for
$N_2O$ predictions: 1) For the region without observation data (normally before day 25), KGML-ag1 predicted stable $N_2O$ fluxes
close to 0 mg N $m^{-2}$ day$^{-1}$ (which is close to the reality in the experiment setting) while GRU caused anomalous peaks of fluxes.
This is because KGML-ag1 has learned knowledge for the whole period from the pretraining process with *ecosys* model
generated synthetic data, but GRU model has no prior knowledge for the period without any data in observations; 2) Although
KGML-ag1 had a lower accuracy than GRU in some chambers, KGML-ag1 can better capture the temporal dynamics of $N_2O$
fluxes compare to GRU, especially when the fluxes are highly variable (e.g. Fig 2 chamber 2).
To validate KGML-ag1 robustness, we further investigated the KGML-ag1 and GRU model performance in different temporal
windows, shrinking from the whole period to the $N_2O$ peak occurrence time (days 1-122, day 30-80, day 40-65 and day 45-60
for year 2016-2018), and performance in $N_2O$ flux, first order gradient of $N_2O$ (slope) and second order gradient of the $N_2O$
(curvature) (Table 2). Slope represents the speed of $N_2O$ flux changes through time and curvature represents the acceleration.
Assessing prediction performance with these two metrics will reveal the model robustness on capture variable dynamics, which
is critical when predicting fast-change variables with hot moments (a short period of time with rare events like flux increasing
quickly) like $N_2O$. First of all, the overall $r^2$ and RMSE of KGML-ag1 for values, slope and curvature were always better than
GRU. In particular, KGML-ag1 captured the peak region (e.g., days 45-60) much better than GRU in both magnitude and
dynamics (Table 2, Fig 2). Even for chamber 2 and 5 in which KGML-ag1 made worse $N_2O$ predictions than GRU ($\Delta r^2$ ranging
from -0.07 to -0.03), it better captured temporal dynamics than GRU in terms of slope ($\Delta r^2$ ranging from 0.08 to 0.16) and
curvature ($\Delta r^2$ from 011 to 0.23) (Table 2). For other chambers, KGML-ag1 outperformed GRU consistently. For chamber 1,
KGML-ag1 had worse $N_2O$ predictions RMSE than GRU but the $\Delta r^2$ increased as the window shrinks to the peak emission
time ($0.07 \rightarrow 0.13$). The slope and curvature for chamber 1 also indicated that KGML-ag1 captured the dynamics much
better than GRU. For chamber 3, KGML-ag1 predicted better $N_2O$ but presented worse slope and curvature RMSE than
GRU (Table 2). However, when explicitly investigating the time series of $N_2O$ flux, slope and curvature in each year, KGML-
ag1 outperformed GRU more significantly in 2017, the year with more complex temporal dynamics of $N_2O$ fluxes, than in

2016 and 2018, especially for chamber 3 (Fig. 2; Fig. S3-4). This investigation supported that KGML-ag1 was more capable for complex dynamics predictions.

Interestingly, the fine-tuned KGML-ag1 model predicted reasonable IMVs including $CO_2$, $NO_3^-$, $NH_4^+$, and VWC with overall $r^2$ of 0.37, 0.39, 0.60, and 0.33 and NRMSE of 0.14, 0.21, 0.09 and 0.18, respectively (Table 3). The time series comparisons between IMV predictions and observations further indicated that KGML-ag1 could reasonably capture both magnitude and dynamics (Fig. 3). KGML-ag2 presented better IMVs predictions than KGML-ag1, with overall $r^2$ of $CO_2$, $NO_3^-$, $NH_4^+$, and VWC increasing by 0.37, 0.17, 0.06 and 0.51, and NRMSE decreasing by 0.05, 0.03, 0.01 and 0.10, respectively, but a slightly lower $r^2$ (decreasing 0.02) of $N_2O$ (Table 3; Fig. S5). This indicated that explicitly simulating each IMV with separated KGML-ag2-IMV modules did not benefit the $N_2O$ flux prediction accuracy, likely due to increasing model complexity which resulted in reduced stability and ignoring the IMV interactions. In addition, we also found all KGML-ag models would perform better by using IMVcb1 (with $CO_2$) than using IMVcb2 (without $CO_2$) in real data tests, indicating feature importance analysis based on synthetic data can be a reasonable substitute for analysis with the often limited real-world data.

**3.3 KGML-ag comparing with other pure ML models**

The results from eight different models showed that KGML-ag1 comparing with other pure ML models consistently provided the lowest RMSE (3.59-3.94 mg N m$^{-2}$ day$^{-1}$, 1.14-1.23 mg N m$^{-2}$ day$^{-2}$, and 0.84-0.89 mg N m$^{-2}$ day$^{-3}$) and highest $r^2$ (0.78-0.81, 0.48-0.56, and 0.23-0.31) for $N_2O$ fluxes, slope and curvature, respectively (Fig. 4). This indicated that KGML-ag1 outperformed other pure ML models in capturing both the magnitude and dynamics of $N_2O$ flux. Meanwhile, we have calculated the uncertainty of mesocosm measurement due to converting hourly data to daily data during 30-80 days by using augmented values minus the mean of the augmented values with lower and upper limits being -10.2 and 10.4 mg N m$^{-2}$ day$^{-1}$, respectively (standard deviation =1.4 mg N m$^{-2}$ day$^{-1}$). KGML-ag1 during the same period has comparable uncertainties based on ensemble simulations with lower and upper limits being -14.4 and 15.2 mg N m$^{-2}$ day$^{-1}$, respectively (calculated by ensemble values minus the mean of ensemble values; standard deviation = 1.3 mg N m$^{-2}$ day$^{-1}$). KGML-ag2 presented slightly better mean scores for $N_2O$ flux predictions than KGML-ag1, but worse scores for slope and curvature and larger uncertainties. This proved the hypothesis discussed in section 3.2 that KGML-ag2 didn't benefit the magnitude and dynamics predictions of $N_2O$ flux with its more complex structure and less connections between IMVs.

Within the tree-based models (DT, RF, GB and XGB), the simplest model DT provided the worst predictions for $N_2O$ flux, slope and curvature. The XGB model provided the highest $N_2O$ flux accuracy with $r^2$ of 0.61-0.63 and RMSE of 5.07-5.17 mg N m$^{-2}$ day$^{-1}$, while the GB model provided best slope and curvature predictions with $r^2$ of 0.38-0.40 and 0.23-0.26, and RMSE of 1.34-1.37 mg N m$^{-2}$ day$^{-2}$ and 0.91-0.95 mg N m$^{-2}$ day$^{-3}$, respectively. The highest $N_2O$ flux accuracy and relatively low slope and curvature accuracy of the XGB model implied that there is a trade-off between the abilities of capturing dynamics and magnitude.


In the group of deep learning models including ANN, GRU and KGML-ag1, ANN provided the worst predictions. Even with
the better $N_2O$ flux predictions than most tree-based models (except XGB), the slope and curvature predictions of ANN were
the worst among all eight models. This implied that the trade-off between accurately capturing $N_2O$ dynamics to magnitude in
ANN was significant. But when considering the temporal dependence, deep learning model GRU and KGML-ag1
outperformed all other models in flux, slope and curvature predictions. This indicated that without considering temporal
dependence the improvement in $N_2O$ flux prediction accuracy could be risky by causing the performance drop in capturing
dynamics.

The detailed model comparisons in each chamber are shown in Fig. 5 ($N_2O$ flux) and Fig. S6-7 ($N_2O$ slope and curvature),
where the results are found to follow the same pattern as described above. In addition, time series comparisons of chamber 3
and 4 in 2017 between different models are presented in Fig. S8 as two examples. For periods without any observed data, we
assumed that the good model predictions should be stable, consistent with the nearest period and close to the reality in the
experiment setting (e.g. no erratic peak and $N_2O$ flux near 0 mg N m$^{-2}$ day$^{-1}$ before day 25). From these comparisons, we infer
that without considering temporal dependence and pretraining process, the tree-based model including DT, RF, GB and XGB
and deep learning model ANN predicted erratic peaks in almost every missing data point, while the GRU model was stable in
short missing period (1-2 days of missing data) and only presented poor performance in long missing period (before day 25).
This improvement by the GRU model may be attributed to the structure of GRU that naturally keeps the historical information
using hidden states, which enables GRU to consider the temporal dependence and make consistent predictions over time.
**3.4 Influence of pretraining process, data augmentation and using IMV initial values as input feature**
After we pretrained the GRU model with synthetic data, the overall $r^2$ of $N_2O$ flux predictions in observed data increased by
0.02, 0.12 and 0.14, and RMSE decreased by 0.23 mg N m$^{-2}$ day$^{-1}$, 0.15 mg N m$^{-2}$ day$^{-2}$ and 0.02 mg N m$^{-2}$ day$^{-3}$ for flux, slope
and curvature predictions, respectively, compared to non-pretrained GRU (No.1-6 in Table 3). The gap between the GRU
model with pretrain and KGML-ag1 in $N_2O$ value prediction shows the improvement resulting from architecture change ($r^2$
increases by 0.01 and RMSE decreases by 0.17 mg N m$^{-2}$ day$^{-1}$). Although pretrained GRU had higher slope and curvature
prediction accuracy than KGML-ag models, it still couldn't achieve the current $N_2O$ value prediction accuracy of KGML-ag1.
Besides, the KGML-ag models had relatively shallow $N_2O$ prediction modules (2-layer GRU KGML-ag-$N_2O$ module of
KGML-ag models vs 4-layer GRU) but included modules for IMV predictions, which therefore increased the model
interpretability.

It's worth noting that prediction accuracy of all KGML-ag models dropped without augmenting the training dataset in the fine-
tuning process (No.7-10 in Table 3). Moreover, the maximum training epochs increased from 800 to 20000, which resulted in
overfitting on the small data set. This indicated that the data augmentation indeed helped the models become more
generalizable and gain better accuracy.

Experiments using zero initial values presented a significant drop in every variable's prediction accuracy (No.11-14 in Table
3). This indicated that the IMV initial values input into the KGML-ag-IMV modules of KGML-ag models influenced not only
the IMV prediction but also the $N_2O$ prediction of the KGML-ag-$N_2O$ module. This shows that there is useful information
transferred from IMVs in the KGML-ag-IMV module to the KGML-ag-$N_2O$ module.
**4 Discussion**
In the previous section, we showed that KGML-ag models can outperform ML models, by invoking architectural constraints
and PB model synthetic data initialization. Compared to traditional PB models such as *ecosys*, KGML-ag models provide
computationally more accurate and efficient predictions (KGML-ag few seconds vs *ecosys* half hour), which is similar to
traditional ML surrogate models (Fig. S9). But KGML-ag goes beyond that by providing more interpretable predictions than
pure ML models.
**4.1 Interpretability of KGML-ag**
The proposed KGML-ag models incorporate causal relations among $N_2O$ related variables/processes as shown in Fig. S10.
Managements, weather forcings and initial values of IMVs influence soil water, soil temperature and soil properties, which
influence the availability of $O_2$ and N as well as the microbe populations in soil, and further influence the nitrification and
denitrification rates. $N_2O$ is produced during both nitrification and denitrification when soil $O_2$ concentration is limited. Our
KGML-ag follows this hierarchical structure by designing KGML-ag-IMV modules representing the soil processes for IMVs
predictions (Fig. 1c-d).

To better explain the time series predictions of $N_2O$ flux (Fig. S1; Fig. 2-3), we separated the observations of each year into
three periods: leading period (before $N_2O$ increasing), increasing period (increasing to the peak) and decreasing period (peak
decreasing to near zero). During the leading period, both $NH_4^+$ and $CO_2$ were increasing immediately in the following few days
following urea N fertilizer application, indicating that urea was decomposing into $NH_4^+$ and $CO_2$ in soil water. With
accumulating $NH_4^+$ in soil, nitrification started producing $NO_3^-$ and consuming $O_2$. $N_2O$ didn't respond to the fertilizer
immediately due to enough $O_2$ in soil. Then when the soil became sufficiently hypoxic, $N_2O$ fluxes entered an increasing
period with $N_2O$ being produced by nitrification and denitrification processes. $CO_2$ fluxes were relatively low and $NH_4^+$ kept
decreasing during this period. Finally, when soil $NH_4^+$ was exhausted and $NO_3^-$ started decreasing due to denitrification, $N_2O$
fluxes then entered the decreasing period. $CO_2$ flux was related to urea decomposition during the leading period, and was more
closely related to $O_2$ demand in other periods. The KGML-ag predictions of $N_2O$ and IMV captured the three periods and
transition points, demonstrating the connections between those variables following the description as above (Fig. 3; Fig. S5).
Although KGML-ag1 obtained lower IMVs prediction accuracy compared to KGML-ag2, it captured the general trends and
was doing better for transitions, especially in $NH_4^+$ predictions. KGML-ag2 overfitted on the observations and ignored the
correlations between IMVs, which resulted in loss in pretrain knowledge, poorer performance in the leading period, and erratic
predictions in the period with missing observations (before day 25).

**4.2 Lessons for KGML-ag development**

The development of KGML-ag in our study is suitable to predict not only $N_2O$ but also other variables, such as $CO_2$, $CH_4$ and
ET, with complicated generation processes relying on the historical states. To develop a capable KGML model, we need to
carefully address three questions:
What kind of ML model is suitable for developing KGML? The answer could be determined by the dominant variation type
of the target variable in the data. If the dominant type is temporal variance, like flux variables in high temporal resolution (e.g.,
daily, or hourly), we should consider ML models with temporal dependency. RNN models such as GRU used in this study,
and CNN models such as casual CNN (Oord et al., 2016) can be good starting ML models. If the dominant type is spatial
variation, like variables in coarse temporal resolution (e.g., monthly or annually) but with high diversity due to soil property,
land cover and climate, we should consider ML models with the ability to deal with edges, hotpoints and categories, such as
CNN;
What physical/chemical constraints can be used to build KGML models? Although physical rules such as mass balance or
energy balance are conceptually straightforward and were proved capable of constraining KGML in predicting lake phosphorus
and temperature dynamics (Hanson et al., 2020; Read et al., 2019), they were excluded in this study according to our
preliminary analysis. The reason is that the mass balance equation of N in the agriculture ecosystem includes too many
unknown and unobservable components such as $N_2$ flux, $NH_3$ flux, N leaching, microbial N, plant N and soil/plant exchange,
which collectively introduce large uncertainties in balance equations and make them hard to be directly applied in the KGML-
ag framework. Other related physical (e.g., diffusion, solution) or chemical (e.g., nitrification, denitrification) processes cannot
be easily added into the KGML-ag structure as rules due to lack of understanding of the process. Instead, as mentioned in Sect.
2.2.4, we used hierarchical structure to enforce an architectural constraint and causal relations among variables, and pretraining
processes to infuse knowledge from *ecosys* to KGML-ag models.
How to involve PB models in the KGML development? An advanced PB model like *ecosys* built upon biophysical and
biochemical rules instead of empirical relations will be a good basis to learn the process, guide the structure and provide the
constraints for KGML. The generated synthetic data in this study helped us get some knowledge about variables such as their
general trends, dynamics and correlations. Such knowledge can be transferred to KGML models from synthetic data in the
pretraining process, which can reduce the efforts to collect large numbers of real-world observation data. Moreover, while
KGML shows great potential beyond PB models, we reckon that equally important for improving $N_2O$ modeling is to continue
improving our understanding of the related processes and mechanisms. Novel data collection and incorporating new
understanding into PB models (e.g., *ecosys*) could provide foundation to further empower KGML (see further discussion in
Sect. 4.3).

**4.3 Limitation and possible improvement**
First, the KGML-ag models in this study are limited by the available observed data. The mesocosm measurements of $N_2O$
fluxes ($16.9\pm11.7$ mg N m$^{-2}$ day$^{-1}$ during days of 45-60; Highest value is 71 mg N m$^{-2}$ day$^{-1}$) and $NO_3^-$ soil concentrations
($59.3\pm20.7$ g N Mg$^{-1}$ during days of 45-60; Highest value is 95.2 g N Mg$^{-1}$) are at the high end of the range that has been
observed by field studies (Fassbinder et al., 2013; Grant et al., 1999, 2006, 2008, 2016; Hamrani et al., 2020;  Venterea et al.,
2011). Some IMVs with high feature importance scores (e.g., $O_2$ flux, $N_2$ flux) or at different depths (e.g., soil $NO_3^-$ at 5 cm
depth, VWC at 5 cm depth), and data out of growing seasons are not included. The direct consequences are that some important
processes cannot be well represented by the current KGML-ag (e.g., $O_2$ demand and N availability for nitrification and
denitrification). Further improvement of KGML should consider three categories of data: target variable $N_2O$ flux, IMVs and
basic inputs (Fig. 1a). For $N_2O$ flux observation, we lack sub-hourly to sub-daily observations to capture the hot moment of
emission during 0-30 days after N fertilizer applications. Besides, the non-growing season can provide 35-65% of the annual
direct $N_2O$ emissions from seasonally frozen croplands and lead to a 17–28 % underestimate of the global agricultural $N_2O$
budget if ignoring its contribution (Wagner-Riddle et al., 2017), but we can barely find observations from non-growing
seasons. For IMVs, we found oxygen demand indicator (e.g., $O_2$ concentration or flux, $CO_2$ flux, $CH_4$ flux), N mass balance
related variables (e.g., $N_2$ flux, soil $NO_3^-$, soil $NH_4^+$, N leaching) and soil water and temperature, can be used to better constrain
the processes and therefore improve the KGML performance. Rohe et al. (2021) also indicated the importance of $O_2$, $CO_2$ and
$N_2$ soil fluxes for $N_2O$ predictions. In addition, the layerwise soil observations (e.g., soil $NO_3^-$, soil VWC) at 0-30 cm depth
can be used to significantly improve the KGML model quality, according to our feature importance analysis (Fig. S2a).
Moreover, continuous monitoring on these variables during the whole year is preferred rather than only during the growing
season, since $N_2O$ flux is largely influenced by previous states. To apply the KGML-ag to large scale, other observational data
including basic inputs of soil/crop properties (e.g., soil bulk density, pH, crop type), management information (e.g., fertilizer,
irrigation, tillage) and weather forcings along with $N_2O$ flux observations are critical for fine-tuning and validating the
developed KGML-ag and therefore explicitly simulating the $N_2O$ or IMVs dynamics under specific conditions. Recent
advances in remote sensing and machine learning have enabled estimating these variables with high-resolution at a large scale
(Peng et al., 2020)

Second, the physical/chemical constraints can be more comprehensive in KGML-ag models. Although current KGML-ag
models are well-initialized with *ecosys* synthetic data and constrained by causal relations of processes with hierarchical
structure, the predicted $N_2O$ flux and IMVs can still violate some basic physical rules like mass balance. As we discussed in
Sec. 4.2, it will be challenging to add physical rules like mass balance equation for N in a complicated agriculture ecosystem
due to data limitations such as missing observations on certain key variables. Using inequalities instead of equations for mass
balance may be one alternative solution. For example, we could use ReLU to add in a limitation for N mass balance residues
which are calculated from known terms not larger than an empirical static value. Besides, better understanding of processes in
the N cycle from fieldworks and lab experiments can also help us design new constraints. This limitation is also partially
related to the data limitation and can be overcomed by involving more complete $N_2O$ data to introduce more powerful
constraints to KGML-ag.

Third, the KGML-ag currently are suffering from dealing with physical/chemical boundary transitions. Boundary transitions
are common in the real world, such as phase change, volume solubility, and soil porosity etc. A detailed PB model generally
coded plenty of "if/else/switch" statements inside to deal with the boundaries. But KGML-ag models based on the GRU are
better at capturing continuous changes, rather than discrete changes. One solution is to include data with boundary information.
In this study, involving IMVs like $O_2$, $CO_2$ and $N_2$, which already have boundary information like water freezing point, N pool
volumes and other complicated boundaries related to soil/crop properties, can significantly improve the model performance.
The data with boundary information could be continuous observation or estimated value from existing data. By using initial
values to predict IMVs, KGML-ag in this study can partially solve the boundary transition problem when observation data is
limited. Another solution is designing new structures of KGML-ag, such as combining ReLU function or including CNN
model which are robust for discrete situations to the RNN models, or designing new constraints to limit the model working
within the thresholds.

Finally, at the current stage we can not claim to have completely opened the black box of KGML-ag, but this framework is a
significant step towards this goal. For example, some ideas implemented in our study, such as using pretraining to transfer
knowledge from a PB model to a ML model, incorporating causal relations by hierarchical structure, predicting IMVs for
tracking middle changes and using initial values as input to reduce data demand, would shed light on the future KGML-ag
framework improvement. Besides, we acknowledge the importance of further testing the KGML-ag over completely
independent datasets, but results presented in this manuscript are sufficient to justify the power of KGML as a framework. The
mesocosm experiment data we used in this study has provided a comprehensive set of inputs and intermediate variables in
addition to the output of $N_2O$ fluxes, thus serving as a unique testbed. We expect to further validate and refine our KGML-ag
model once more gold standard data of $N_2O$ fluxes along with other relevant inputs and intermediate variables become publicly
available. Moreover, incorporating more and more domain knowledge into KGML-ag will be possible for further
improvement, but we don't think KGML-ag will become inefficient as it becomes more like the PB model. In fact, to efficiently
emulate components of PB models has been proposed as a research frontier in hybrid modeling for earth system science
(Reichstein et al., 2019; Irrgang et al., 2021), with latest advances occurring in weather forecasts (Bauer et al., 2021). By using
a hybrid model, computationally inefficient components of PB can be identified one by one, and be replaced with more efficient
ML-based surrogates to eventually obtain the most efficient model. Further KGML-ag model development will also need to
balance efficiency, accuracy and interpretability.

## 5 Conclusions

In this study, two KGML-ag models have been developed, validated, and tested for agricultural soil $N_2O$ flux prediction using
synthetic data generated by the PB model *ecosys* and observational data from a mesocosm facility. The results show that
KGML-ag models can outperform PB and pure ML models in $N_2O$ prediction in not only magnitude (KGML-ag1 $r^2 = 0.81$ vs
best ML model GRU $r^2 = 0.78$) but also dynamics (KGML-ag1 accuracy minus GRU accuracy, slope $\Delta r^2 = 0.06$ and curvature
$\Delta r^2 = 0.08$). KGML-ag can also defeat the PB model *ecosys* in efficiency by completing *ecosys*'s half-hour job within a few
seconds. Compared to ML models, KGML-ag models can better represent complex dynamics and high peaks of $N_2O$ flux.
Moreover, with IMV predictions and hierarchical structures, KGML-ag models can provide biogeophysical/chemical
information about key processes controlling $N_2O$ fluxes, which will be useful for interpretable forecasting and developing
mitigation strategies. Data demand for the KGML-ag models is significantly reduced due to involving IMV initial values and
pretrain processes with synthetic data. This study demonstrated that the potential of KGML-ag application in the complex
agriculture ecosystem is high and illustrates possible pathways of KGML-ag development for similar tasks. Further
improvement of our KGML-ag models can involve general principles to further constrain the predictions through loss functions
or architectures, but call for more detailed, high temporal resolution $N_2O$ observation data from field measurements.

## Code and Data Availability

The code and data used in this study can be found at https://doi.org/10.5281/zenodo.5504533.

## Author contributions

LL and ZJ conceived the study. WZ and YY conducted *ecosys* simulations and provided synthetic data. LL and SX processed
the data and developed the KGML-ag model. LL, SX and SW carried the experiments out with supervisions from ZJ, JT, KG,
and VK. TJG, MDE, ALF and LTM shared mesocosm observations and interpreted the data. LL wrote the first draft of the
manuscript with further editing from TK on figures and tables. ZJ, SX, JT, KG, XJ, BP, YY, WZ and VK further edited the
manuscript.

## Competing interests

The authors declare that they have no conflict of interest.

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

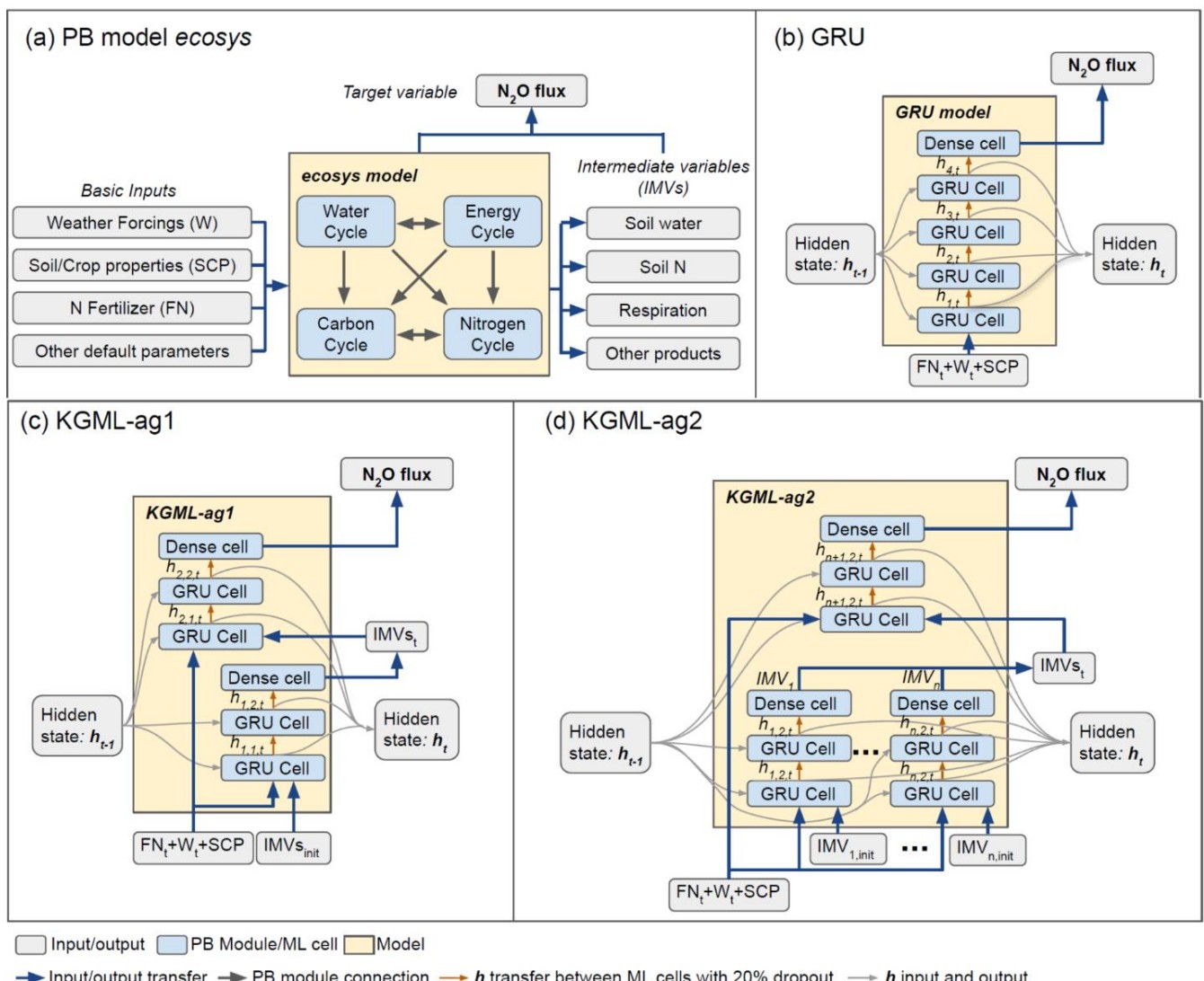

**Figure 1: The model structures. a) The *ecosys* model; b) Gated recurrent unit (GRU) model; c) KGML-ag1 model with a hierarchical structure; d) KGML-ag2 model with a hierarchical structure with separated GRU modules for IMV predictions. Specifically, in our KGML model design, weather forcings (W) include temperature (TMAX, TDIF), precipitation (PRECN), radiation (RADN), humidity (HMAX and HDIF) and wind speed (WIND); soil/crop properties (SCP) include bulk density (TBKDS), sand content (TCSAND), silt content (TCSILT), pH (TPH), cation exchange capacity (TCEC), soil organic carbon (TSOC), planting day of the year (PDOY) and crop type (CROPT); IMVs include $CO_2$ flux, soil $NO_3^-$ concentration, soil $NH_4^+$ concentration, and soil volumetric water content (VWC).**

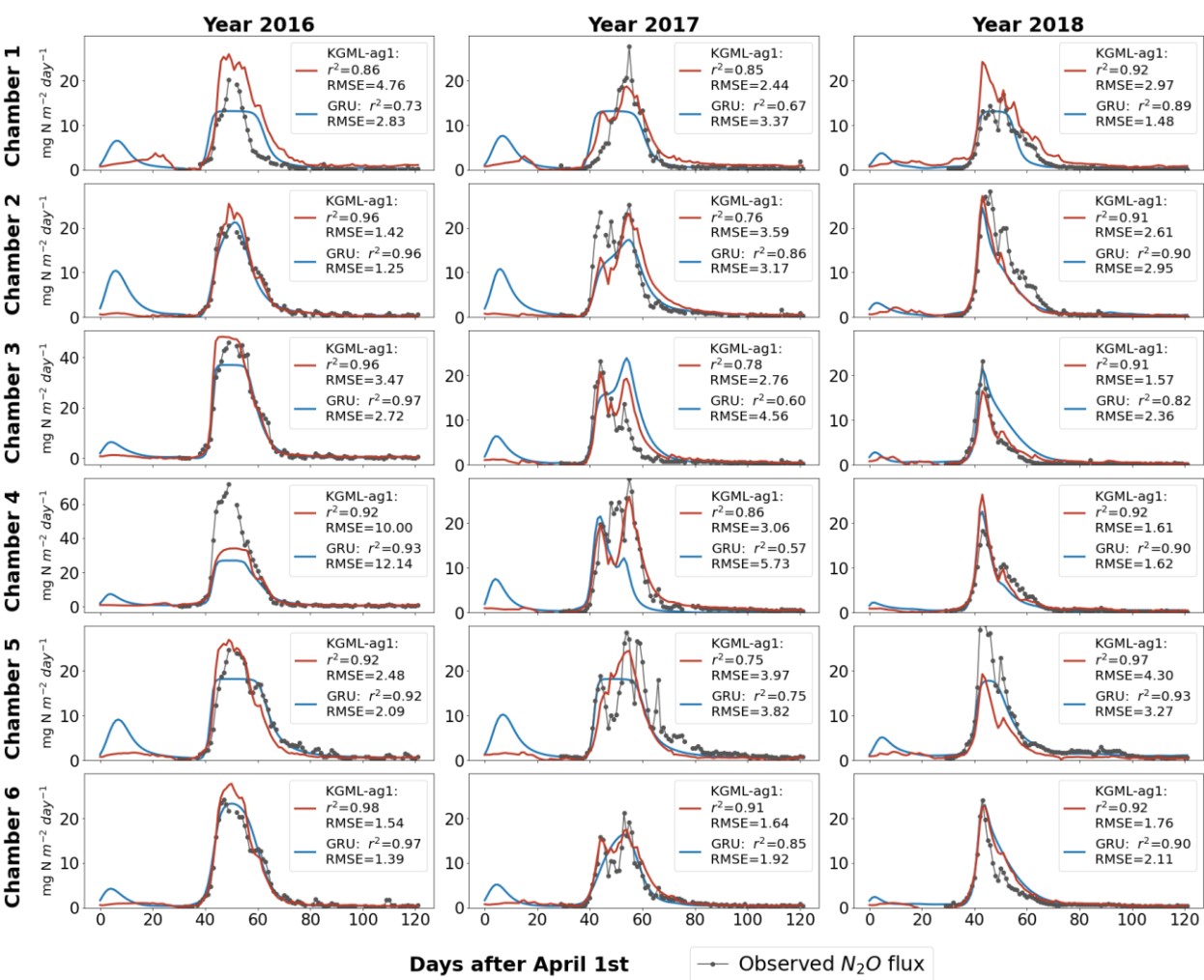

747

Figure 2: Leave-one-out cross validation of time series of N₂O flux (mg N m⁻² day⁻¹) predicted by the pure non-pretrained GRU
model (blue line) and KGML-ag1 model (red line). Observations are shown as black line-dots. Validation results for each chamber
were based on out-of-sample predictions by models trained by other five chambers.

751

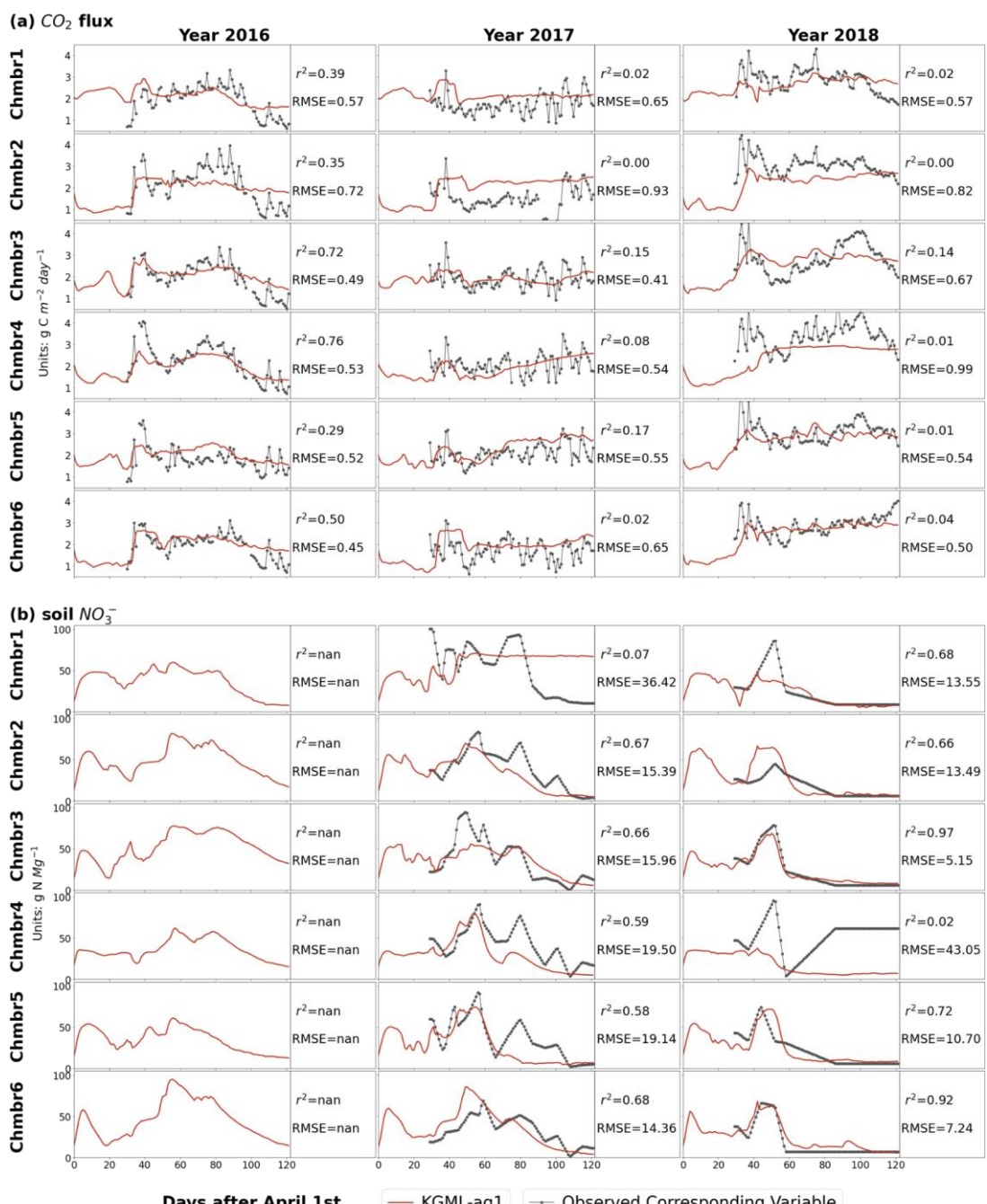

**Figure 3: Leave-one-out cross validation of time series of IMVs predicted by KGML-ag1 model (red line). Observations are shown as black line-dots. Validation results for each chamber were based on out-of-sample predictions by models trained by other five chambers. Chmb is the abbreviation for chamber. $r^2$ and RMSE are calculated and present in each year and chamber. The $CO_2$ flux and soil $NO_3^-$ concentration units are g C m$^{-2}$ day$^{-1}$ and g N Mg$^{-1}$, respectively.**

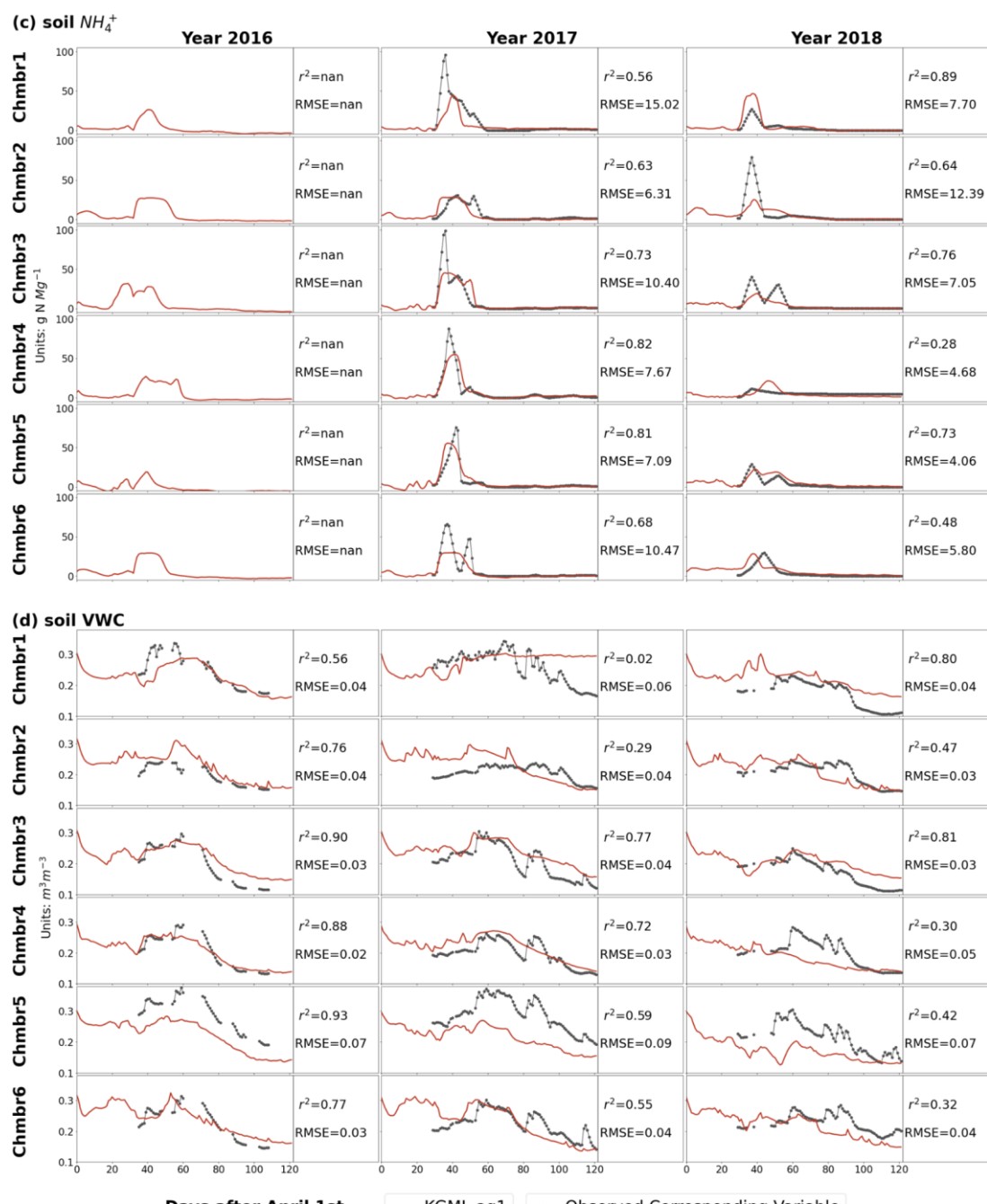

Figure 3 Contd.: Leave-one-out cross validation of time series of IMVs predicted by KGML-ag1 model (red line). Observations are shown as black line-dots. Validation results for each chamber were based on out-of-sample predictions by models trained by other five chambers.Chmb is the abbreviation for chamber. r² and RMSE are calculated and present in each year and chamber. The soil NH₄⁺ concentration and soil VWC units are g N Mg⁻¹ and m³ m⁻³, respectively.

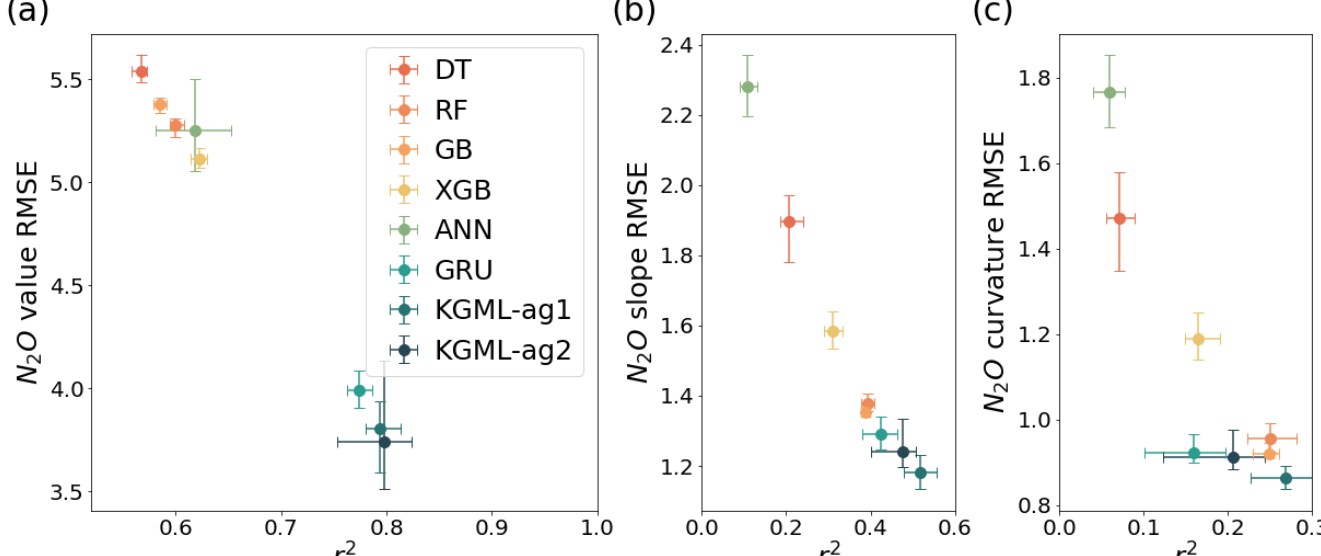

764

**Figure 4: The comparisons of overall prediction accuracy from leave-one-out cross validation for N₂O value (a), 1st order gradient (slope, b) and 2nd order gradient (curvature, c) between four tree-based ML models (DT, RF, GB and XGB), two deep learning models (ANN and GRU) and KGML-ag models. The overall performances were calculated by comparing out-of-sample predictions (each chamber's predictions were from models trained by other five chambers) from all validated chambers with observations. Different color symbols represent the different models. The x- and y-error bars are coming from the maximum and minimum scores of ensemble experiments. The dot represents the mean score of the ensemble experiments.**

771

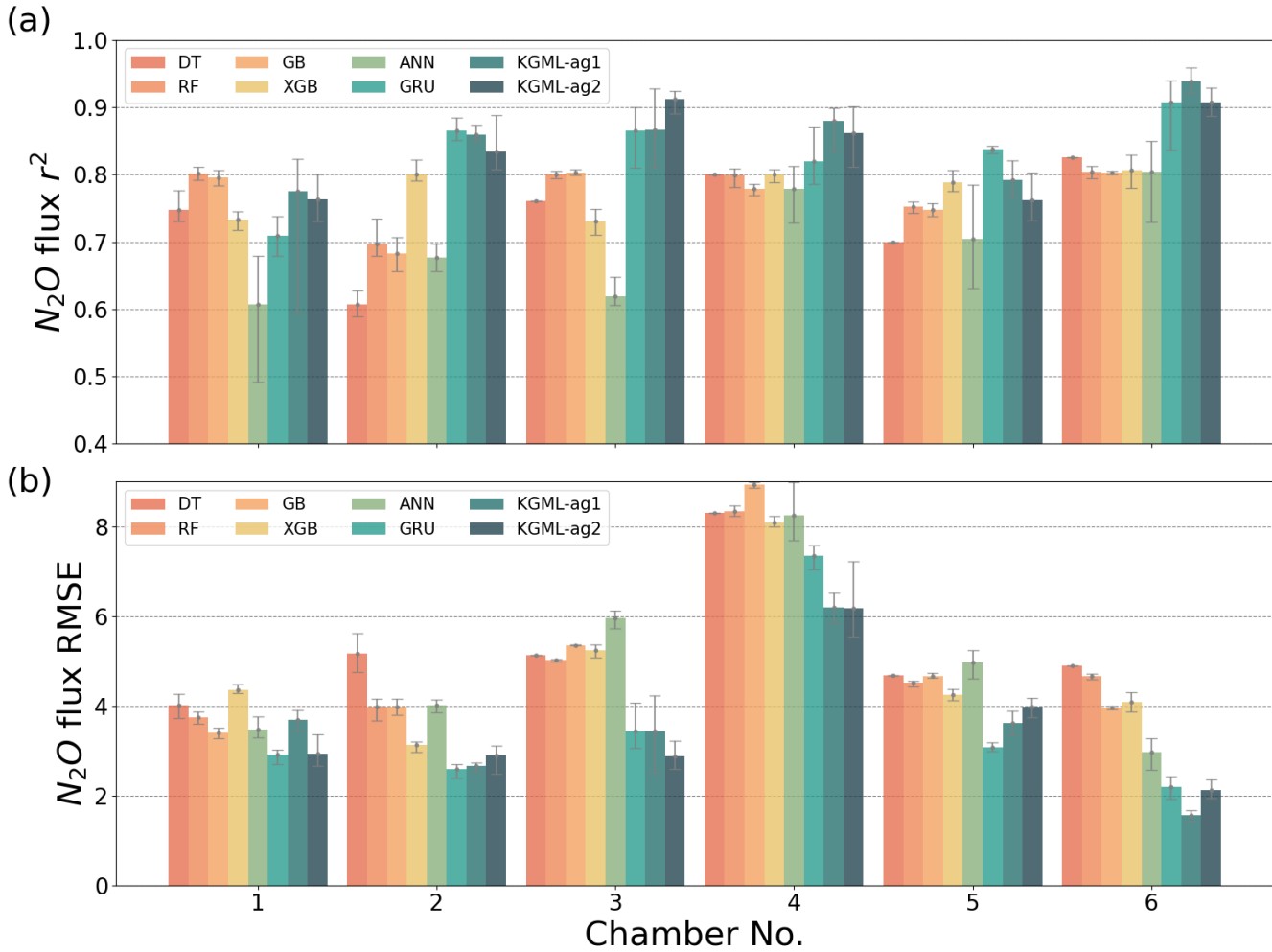

Figure 5: The comparisons of N₂O flux prediction accuracy r² (a) and (b) RMSE from leave-one-out cross validation, between four tree-based ML models (DT, RF, GB and XGB), two deep learning models (ANN and GRU) and KGML-ag models in six chambers. Validation results for each chamber were based on out-of-sample predictions by models trained by other five chambers. The gray error bars are coming from the maximum and minimum scores of ensemble experiments.

Table 1: Pretrain results for different model and IMV combinations using *ecosys* synthetic data. Only performances from testing data sets (synthetic data from 19 counties) were presented.

| No. | Pretrain Model | Input Feature N | $N_2O$ r² | $N_2O$ RMSE | $CO_2$ r² | $CO_2$ NRMSE | $NO_3^-$ r² | $NO_3^-$ NRMSE | $NH_4^+$ r² | $NH_4^+$ NRMSE | VWC r² | VWC NRMSE |
|---|---|---|---|---|---|---|---|---|---|---|---|---|
| 1 | GRU+76IMVs | 76 IMVs+FN+7Ws+8SCP | 0.98 | 0.54 | --[a] | -- | -- | -- | -- | -- | -- | -- |
| 2 | GRU+IMVcb1 | 4 IMVs+FN+7Ws+8SCP | 0.92 | 1.15 | -- | -- | -- | -- | -- | -- | -- | -- |
| 3 | GRU+IMVcb2 | 3 IMVs+FN+7Ws+8SCP | 0.90 | 1.26 | -- | -- | -- | -- | -- | -- | -- | -- |
| 4 | GRU | FN+7Ws+8SCP | 0.89 | 1.37 | -- | -- | -- | -- | -- | -- | -- | -- |
| 5 | KGML-ag1+IMVcb1_ini | FN+7Ws+8SCP+4IMV_ini | 0.90 | 1.24 | 0.91 | 0.06 | 0.95 | 0.03 | 0.98 | 0.03 | 0.95 | 0.04 |
| 6 | KGML-ag1+IMVcb2_ini | FN+7Ws+8SCP+3IMV_ini | 0.90 | 1.26 | -- | -- | 0.94 | 0.03 | 0.97 | 0.03 | 0.95 | 0.04 |
| 7 | KGML-ag2+IMVcb1_ini | FN+7Ws+8SCP+4IMV_ini | 0.90 | 1.27 | 0.92 | 0.05 | 0.95 | 0.02 | 0.98 | 0.03 | 0.96 | 0.04 |
| 8 | KGML-ag2+IMVcb2_ini | FN+7Ws+8SCP+3IMV_ini | 0.91 | 1.19 | -- | -- | 0.95 | 0.00 | 0.99 | 0.02 | 0.95 | 0.04 |

[a]The empty slot indicates that the model does not predict that variable.

Table 2: Prediction accuracy comparisons between non-pretrained GRU model and KGML-ag1.

| | No. | $N_2O$, KGML-ag1 minus GRU — All time[b] | Day 30-80 | Day 40-65 | Day 45-60 | $N_2O$ 1st order gradient, KGML-ag1 minus GRU — All time | Day 30-80 | Day 40-65 | Day 45-60 | $N_2O$ 2nd order gradient, KGML-ag1 minus GRU — All time | Day 30-80 | Day 40-65 | Day 45-60 |
|---|---|---|---|---|---|---|---|---|---|---|---|---|---|
| $\Delta r^{2\,a}$ | All data | 0.03[c] | 0.04 | 0.07 | 0.10 | 0.07 | 0.07 | 0.07 | 0.15 | 0.08 | 0.08 | 0.09 | 0.11 |
| | Chamber1 | 0.07 | 0.10 | 0.20 | 0.13 | 0.18 | 0.18 | 0.19 | 0.14 | 0.08 | 0.09 | 0.09 | 0.02 |
| | Chamber2 | **-0.04** | **-0.05** | **-0.07** | **-0.05** | 0.08 | 0.09 | 0.09 | 0.16 | 0.20 | 0.20 | 0.20 | 0.23 |
| | Chamber3 | 0.06 | 0.06 | 0.08 | 0.06 | 0.04 | 0.04 | 0.04 | 0.13 | **-0.01** | **-0.01** | **-0.01** | 0.07 |
| | Chamber4 | 0.06 | 0.08 | 0.12 | 0.07 | 0.05 | 0.05 | 0.05 | 0.14 | 0.07 | 0.07 | 0.08 | 0.12 |
| | Chamber5 | **-0.05** | **-0.06** | **-0.07** | **-0.03** | 0.09 | 0.09 | 0.10 | 0.16 | 0.13 | 0.13 | 0.15 | 0.11 |
| | Chamber6 | 0.03 | 0.04 | 0.08 | 0.17 | 0.14 | 0.14 | 0.15 | 0.22 | 0.12 | 0.13 | 0.14 | 0.23 |
| $\Delta RMSE^a$ | All data | -0.41 | -0.56 | -0.84 | -1.19 | -0.07 | -0.10 | -0.14 | -0.20 | -0.03 | -0.05 | -0.07 | -0.08 |
| | Chamber1 | **0.80** | **1.06** | **1.21** | **1.70** | 0.00 | 0.00 | -0.02 | 0.00 | **0.05** | **0.07** | **0.10** | **0.18** |
| | Chamber2 | **0.08** | **0.11** | **0.07** | -0.04 | -0.10 | -0.13 | -0.18 | -0.14 | -0.10 | -0.14 | -0.19 | -0.22 |
| | Chamber3 | -0.71 | -0.96 | -1.30 | -2.09 | **0.03** | **0.04** | **0.07** | **-0.25** | **0.09** | **0.13** | **0.17** | **0.08** |
| | Chamber4 | -1.68 | -2.27 | -3.09 | -3.81 | -0.11 | -0.15 | -0.21 | -0.26 | -0.05 | -0.07 | -0.09 | -0.16 |
| | Chamber5 | **0.53** | **0.69** | **0.86** | **0.99** | -0.10 | -0.14 | -0.20 | -0.23 | -0.09 | -0.12 | -0.18 | -0.14 |
| | Chamber6 | -0.20 | -0.27 | -0.37 | -0.61 | -0.14 | -0.20 | -0.29 | -0.33 | -0.07 | -0.10 | -0.15 | -0.19 |

[a]Leave-one-out cross validation results for each chamber were based on out-of-sample predictions by models trained by other five chambers. The "All data" performances were calculated by comparing out-of-sample predictions from all validated chambers with observations. The difference of r² ($\Delta r^2$), and difference of RMSE ($\Delta RMSE$, units are mg N m$^{-2}$ day$^{-1}$, mg N m$^{-2}$ day$^{-2}$, mg N m$^{-2}$ day$^{-3}$ for $N_2O$ value, 1st order gradient and 2nd order gradient, respectively) were calculated by values from KGML-ag1 minus values from GRU.

[b]Results from different time windows of different chambers during the period of April 1st-July31st (Days1-122) were detected.



788 cNone bold values mean KGML-ag1 outperforms GRU, while **bold** values mean the opposite.

**Table 3: Experiments for measuring GRU and KGML-ag models performance, and influence of pretraining process, training data**
**augmentation and IMV initial values.**

| No. | Retrain Model | Experiment | $N_2O$ | | $N_2O$ 1st order gradient | | $N_2O$ 2nd order gradient | | $CO_2$ | | $NO_3^-$ | | $NH_4^+$ | | VWC | |
|---|---|---|---|---|---|---|---|---|---|---|---|---|---|---|---|---|
| | | | $r^{2\,c}$ | RMSE[c] | $r^2$ | RMSE | $r^2$ | RMSE | $r^2$ | NRMSE | $r^2$ | NRMSE | $r^2$ | NRMSE | $r^2$ | NRMSE |
| 1 | **GRU, baseline[a]** | **No Pretrain** | **0.78** | **4.00** | **0.45** | **1.27** | **0.20** | **0.90** | --[b] | -- | -- | -- | -- | -- | -- | -- |
| 2 | GRU | Pretrain | 0.80 | 3.77 | 0.57 | 1.12 | 0.34 | 0.82 | -- | -- | -- | -- | -- | -- | -- | -- |
| 3 | KGML-ag1+ IMVcb1_ini | Original setting | 0.81 | 3.60 | 0.51 | 1.20 | 0.28 | 0.87 | 0.37 | 0.14 | 0.39 | 0.21 | 0.60 | 0.09 | 0.33 | 0.18 |
| 4 | KGML-ag1+ IMVcb2_ini | Original setting | 0.80 | 3.71 | 0.49 | 1.22 | 0.21 | 0.91 | -- | -- | 0.37 | 0.22 | 0.53 | 0.10 | 0.33 | 0.19 |
| 5 | KGML-ag2+ IMVcb1_ini | Original setting | 0.79 | 3.77 | 0.48 | 1.23 | 0.22 | 0.90 | 0.74 | 0.09 | 0.46 | 0.18 | 0.66 | 0.08 | 0.84 | 0.08 |
| 6 | KGML-ag2+ IMVcb2_ini | Original setting | 0.78 | 3.91 | 0.47 | 1.24 | 0.20 | 0.91 | -- | -- | 0.49 | 0.18 | 0.69 | 0.08 | 0.84 | 0.08 |
| 7 | KGML-ag1+ IMVcb1_ini | No augmentation | 0.80 | 3.73 | 0.49 | 1.22 | 0.22 | 0.90 | 0.38 | 0.14 | 0.38 | 0.21 | 0.61 | 0.09 | 0.37 | 0.17 |
| 8 | KGML-ag1+ IMVcb2_ini | No augmentation | 0.77 | 4.04 | 0.41 | 1.31 | 0.13 | 0.95 | -- | -- | 0.38 | 0.21 | 0.53 | 0.10 | 0.35 | 0.18 |
| 9 | KGML-ag2+ IMVcb1_ini | No augmentation | 0.76 | 4.06 | 0.45 | 1.27 | 0.16 | 0.95 | 0.69 | 0.10 | 0.21 | 0.25 | 0.60 | 0.09 | 0.80 | 0.09 |
| 10 | KGML-ag2+ IMVcb2_ini | No augmentation | 0.74 | 4.27 | 0.48 | 1.23 | 0.21 | 0.90 | -- | -- | 0.40 | 0.21 | 0.60 | 0.09 | 0.81 | 0.09 |
| 11 | KGML-ag1+ IMVcb1_ini | Zero initial values | 0.48 | 6.27 | 0.26 | 1.49 | 0.08 | 1.00 | 0.19 | 0.16 | 0.25 | 0.25 | 0.47 | 0.12 | 0.14 | 0.25 |
| 12 | KGML-ag1+ IMVcb2_ini | Zero initial values | 0.49 | 5.94 | 0.31 | 1.41 | 0.13 | 0.95 | -- | -- | 0.31 | 0.25 | 0.38 | 0.13 | 0.24 | 0.25 |
| 13 | KGML-ag2+ IMVcb1_ini | Zero initial values | 0.48 | 6.05 | 0.12 | 1.66 | 0.01 | 1.09 | 0.58 | 0.12 | 0.34 | 0.25 | 0.21 | 0.13 | 0.56 | 0.31 |
| 14 | KGML-ag2+ IMVcb2_ini | Zero initial values | 0.39 | 6.60 | 0.15 | 1.59 | 0.04 | 1.01 | -- | -- | 0.16 | 0.27 | 0.27 | 0.12 | 0.53 | 0.31 |

aNo.1-6 includes the experiments with original simulation settings as described in Sec. 2 and bold values refers to the baseline GRU
simulation; No.7-10 include the experiments without data augmentation during the finetuning process; And No. 11-14 includes the
experiments of replacing original IMV initial values with zeros.
bThe empty slot indicates that the model does not predict that variable.
cThe leave-one-out cross validation overall performances were calculated by comparing out-of-sample predictions (each chamber's
predictions were from models trained by other five chambers) from all validated chambers with observations.