# Peer review of "Learning to Simulate Agroecosystems: A Case Study of Estimating"

_Geoscientific Model Development, 2021_

## Author Comment (AC3)

**Response Letter**

We are grateful to all comments and suggestions from three reviewers and have carefully addressed their concerns point by point. Major changes include:

(1) We have conducted uncertainty analysis for all pure machine learning models and KGML models presented in out study to include the machine learning model uncertainties;

(2) The uncertainties of process-based model *ecosys* and its performance over various ecosystem for $N_2O$ and $CO_2$ have been added into the maintext;

(3) We have added LSTM results into the supplement and comparing with all other models for reference;

(4) We have added a new paragraph in discussion to address the concerns of KGML-ag limitations;

(5) We have clarified all the confusing parts which have been pointed out by reviewers.

By changing these major concerns and many other minor comments and suggestions, we believe the quality of this manuscript is improved. Below, please find our detailed responses point-by-point.

Please be aware of the formatting of all responses:
1. Reviewer comment in **black**, response in **blue** and quotation from the main text in ***red***;
2. The line number is based on the clean version of the revised manuscript, not the track change version.

**To Reviewer 1**

Liu et al. presented a promising predictive framework that combined a process-based model (physical knowledge and pre-train dataset) and a machine learning model for agroecosystem $N_2O$ emission estimate. The modeling framework is robust and thoroughly validated. This work will be an important milestone towards a better understanding, monitoring, and predicting agroecosystem greenhouse gas emissions.

The paper is well organized and written. Below are some of my comments that may help elucidate the strength and limitations of the proposed KGML-ag framework.

Response: We really appreciate that the reviewer recognized our efforts in developing the proper knowledge guided machine learning framework for agroecosystem. To improve the quality of this study, we have carefully revised the manuscript based on the reviewer's comments and suggestions shown as below:

1. Robustness of physical (prior) knowledge

*ecosys* model plays a central role in guiding the ML model in terms of structure and providing a pre-train dataset. It will be important to discuss the structure uncertainty in *ecosys* $N_2O$ module, including e.g., underlying theories, major processes, difference/similarity to the classic leaky pipe type model (Davidson et al., 2000), and so on.

Reference:
Davidson, E. A., Keller, M., Erickson, H. E., Verchot, L. V., & Veldkamp, E. (2000). Testing a conceptual model of soil emissions of nitrous and nitric oxides: using two functions based on soil nitrogen availability and soil water content, the hole-in-the-pipe model characterizes a large fraction of the observed variation of nitric oxide and nitrous oxide emissions from soils. Bioscience, 50(8), 667-680.

Response: Thank you so much for this suggestion. In this revision, we have added a detailed description on the major processes of $N_2O$ production and transfer in *ecosys* model, and on the differences between traditional pipeline $N_2O$ model and *ecosys* model. You can find the description in the manuscript section 2.2.1 (from Line xxx to yyy) as:

"It represents $N_2O$ evolution in the microbe-engaged processes of nitrification-denitrification using substrate kinetics that are sensitive to soil nitrogen availability, soil temperature, soil moisture, and soil oxygen status (Grant and Pattey 2008). Two groups of microbial populations, autotrophic nitrifiers and heterotrophic denitrifiers, produce $N_2O$ with specific competitive or cooperative relations in *ecosys* when $O_2$ availability fails to meet $O_2$ demand for their respirations and $NO_2^-$ become alternative electron acceptors. $N_2O$ transfer within soil layers and from soil to the atmosphere is driven by concentration gradient using diffusion-convection-dispersion equations, in the forms of gaseous and aqueous $N_2O$ under control of volatilization-dissolution (Grant et al., 2016). Unlike the pipeline model described by Davidson et al. (2000) , which mainly consider the correlations of $N_2O$ production with nitrogen availability and of $N_2O$ emitting with soil water content, *ecosys* enables integrative effects of energy, water, nitrogen availability on $N_2O$ production and $N_2O$ transfer via the microbial population dynamics and their

interactions with soil, plant, and atmospheric dynamics, under diverse meteorological and anthropogenic disturbances (e.g. runoff, drainage, tillage, irrigation, soil erosion)."

Again ecosys provides pretrain dataset, which has its own uncertainty and biases. It's worthwhile to at least show some ecosys model performance across various different conditions at agroecosystems. For example, does ecosys pick up the high-frequency signals (fluctuation) of CO2/N2O flux that are observed in the chambers data? If not, is that the reason why PGML-ag could not capture the high fluctuation of CO2/N2O emissions in the field?

Response: We really appreciate this comment which suggests to show the capability of *ecosys* model as the domain knowledge provider. To show the *ecosys* model performance on simulation of $CO_2$ and $N_2O$ emissions at field, we have added detailed quantitative comparisons between model simulations and observations in the manuscript section 2.2.1 (from line xxx to yyy):

"The performances of *ecosys* on $CO_2$ and $N_2O$ fluxes have been proven by numerous field-level and regional studies, including net ecosystem $CO_2$ exchange (NEE, $R^2 = 0.87$) and leaf area index (LAI, $R^2 = 0.78$) from six flux towers in the Midwest, USDA census reported corn yield ($R^2 = 0.83$) and soybean yield ($R^2 = 0.80$), satellite-derived GPP for corn ($R^2 = 0.83$) and soybean ($R^2 = 0.85$) from Illinois, Iowa and Indiana in the US Midwest, and cumulative $N_2O$ emissions ($R^2 = 0.36$) across eight Midwestern states (Wang et al., 2021; Yang et al., 2022)."

If you are interested in the more detailed performance of field level $N_2O$ emission simulation using *ecosys* model, you may review 1) the papers of Grant et al (2006, 2008) to find the influences of fertilizer rate and temperature on $N_2O$ emissions in fertilized agriculture soil; 2) the paper of Grant et al (1999) to find the influences of spring thawing; and 3) the papers of Grant et al (2010, 2016) to check the $N_2O$ simulation performances at managed forest and grassland.

2. It's not obvious which variables are used as inputs or intermediate variables and how that relates to the feature importance ranking. It will be better to show each variable in Figure 1. For example, W will be temperature and precipitation. Furthermore, feature importance analysis highlight NH3, H2, N2, O2, CH4, ET, CO2 are important variables that drive N2O emission (~ L230). It's not clear in the main text, how this feature importance ranking helps the design of PGML-ag. What can we get out of this feature importance analysis?

Response: Thanks for pointing out the confusing part of how feature importance related to KGML model development. In this revision, we have extended descriptions in Figure 1 caption to explain W, SCP and IMVs that are used in our study.

"Figure 1: The model structures. a) The *ecosys* model; b) Gated recurrent unit (GRU) model; c) KGML-ag1 model with a hierarchical structure; d) KGML-ag2 model with a hierarchical structure using separated GRU modules for IMV predictions. Specifically, in our KGML model design, weather forcings (W) include temperature (TMAX, TDIF), precipitation (PRECN), radiation (RADN), humidity (HMAX and HDIF) and wind speed (WIND); soil/crop properties (SCP) include bulk density (TBKDS), sand content (TCSAND), silt content (TCSILT), pH (TPH), cation exchange capacity (TCEC), soil organic

carbon (TSOC), planting day of the year (PDOY) and crop type (CROPT); IMVs include $CO_2$ flux, soil $NO_3^-$ concentration, soil $NH_4^+$ concentration, and soil volumetric water content (VWC)."

Feature importance analysis was the first step in our study to learn the knowledge from synthetic data generated by the *ecosys* model and to investigate the correlation between input/intermediate variables and $N_2O$ fluxes. The importance rankings help us to put low/median/high attention to available variables during model development (e.g. $CO_2$ was tested as a higher ranking variable than others so that we paid high attention to it by testing two different combinations of IMVs w/o $CO_2$). In addition, the rankings will provide guidance of future $N_2O$ related measurement, which is discussed in section 4.3. We have revised paragraph two in section 2.2.4 to highlight how feature importance rankings help our model development (from line xxx to yyy).

"Variables ranked high in feature importance analysis should be primarily considered during model development. To develop a functionable KGML-ag, we further investigated the feature importance of four IMVs that are available from mesocosm observations including $CO_2$, $NO_3^-$, VWC and $NH_4^+$, which were ranked 7th, 20th, 58th, 60th respectively in 92 input features of synthetic data (Fig. S2a). We used these four available IMVs to create two input combinations: 1) $CO_2$ flux, $NO_3^-$, VWC and $NH_4^+$ (IMVcb1), and 2) $NO_3^-$, VWC and $NH_4^+$ (IMVcb2). The objective of building IMVcb2 was to investigate the importance of highly ranked variable $CO_2$ flux (by removing it from the inputs), and the impact of mixing-up flux and non-flux variables on model performance. "

3.  There is a lack of discussion on uncertainty in PGML-ag, which is fundamentally important for predictive modeling. Also, what about chamber measurements uncertainty?

Response: Thank you for pointing out this concern for predictive modeling. To address the uncertainty of the machine learning models and KGML-ag model, we have conducted 10 ensemble experiments for different model structures (DT, RF, GB, XGB, ANN, GRU, KGML-ag1 and KGML-ag2). Corresponding method part in section 2.1 has been updated (from line xxx to yyy).

"We further benchmarked KGML-ag models and uncertainties with other pure ML models without considering temporal dependence, including Decision Tree (DT), Random Forest (RF), Gradient Boosting (GB) from the sklearn package (https://scikit-learn.org/stable/), Extreme Gradient Boosting (XGB) from the XGBoost package (https://xgboost.readthedocs.io/en/latest/) and a 6-linear-layer artificial neural network (ANN) with the mesocosm experiment data by 10 times ensemble experiments (Fig. 4-5; Fig. S6-8);"

The new results have been updated in Figure 4 and Figure 5 (also as Figure R1 and R2 below) in the main text and Figure S6-S7 (also as Figure R3 and R4 below) in the supplementary. We have also updated values in section 3.3 accordingly. For chamber measurement uncertainty, we have cited the original thesis (Miller L., 2021) including the mesocosm experiment settings, instruments and related measurement uncertainties (e.g. Figure 2.2 in the thesis). In our study, we also used a data augmentation method to cover the uncertainties caused by converting hourly observations to daily observations. The data augmentation method has been described in section 2.2.2 paragraph 3.

[Figure]

Figure R1: The comparisons of overall prediction accuracy for $N_2O$ value (a), 1st order gradient (slope, b) and 2nd order gradient (curvature, c) between four tree-based ML models (DT, RF, GB and XGB), two deep learning models (ANN, GRU), and KGML-ag models. Different color symbols represent the different models. The x- and y-error bars are coming from the maximum and minimum scores of ensemble experiments. The dot represents the mean score of the ensemble experiments.

[Figure]

Figure R2: The comparisons of $N_2O$ flux prediction accuracy $r^2$ (a) and (b) RMSE, between four tree-based ML models (DT, RF, GB and XGB), two deep learning models (ANN and GRU), and KGML-ag models in 6 chambers. The gray error bars are coming from the maximum and minimum scores of ensemble experiments.

[Figure]

Figure R3: The comparisons of N2O 1st order gradient prediction accuracy r2 (a) and (b) RMSE, between four tree-based ML models (DT, RF, GB and XGB), two deep learning models (ANN and GRU) and KGML-ag models in 6 chambers. The gray error bars are coming from the maximum and minimum scores of ensemble experiments.

[Figure]

Figure R4: The comparisons of N$_2$O 2nd order gradient prediction accuracy r$^2$ (a) and (b) RMSE, between four tree-based ML models (DT, RF, GB and XGB), two deep learning models (ANN and GRU) and KGML-ag models in 6 chambers. The gray error bars are coming from the maximum and minimum scores of ensemble experiments.

L254 based on the structure of process representation in ecosys

Response: We have revised the sentence based on your suggestion (Line xxx to yyy).

"We built a hierarchical structure based on the structure of process representation in *ecosys* to first predict IMVs and then simulate N$_2$O with predicted IMVs;"

References:

Grant, R. F., Black, T. A., Jassal, R. S., & Bruemmer, C.: Changes in net ecosystem productivity and greenhouse gas exchange with fertilization of Douglas fir: Mathematical modeling in *ecosys*. Journal of Geophysical Research: Biogeosciences, 115(G4), 2010.

Grant, R. F., & Pattey, E.: Mathematical modeling of nitrous oxide emissions from an agricultural field during spring thaw. Global Biogeochemical Cycles, 13(2), 679-694, 1999.

Grant, R. F., & Pattey, E.: Temperature sensitivity of $N_2O$ emissions from fertilized agricultural soils: Mathematical modeling in ecosys. Global biogeochemical cycles, 22(4), 2008.

Grant, R. F., Neftel, A., & Calanca, P.: Ecological controls on $N_2O$ emission in surface litter and near-surface soil of a managed grassland: modelling and measurements, Biogeosciences, 13(12), 3549-3571, 2016.

Grant, R. F., Pattey, E., Goddard, T. W., Kryzanowski, L. M., & Puurveen, H.: Modeling the effects of fertilizer application rate on nitrous oxide emissions, Soil Science Society of America Journal, 70(1), 235-248, 2006.

**To Reviewer 2**

General comments

This manuscript presents a new method for estimating N2O flux from cropland. The inputs to the method are known fertilization rate, weather forcings, soil and crop properties. The method also requires initial concentrations of nitrate ions, ammonium ions, and water in the soil, and optionally CO2 flux. The method employs gated recurrent networks organized in a hierarchical structure to mirror the time-dependence and causality present in the process. A process-based model provides pre-training data, and fine-tuning is done using observations from mesocosm experiments. The trained neural network models outperform the process-based model and many basic machine learning approaches.

The methodology employed is both novel and sound. The use of GRUs in hierarchical structures is well-justified and appropriate to the problem. The models have been well-validated, and various alternate choices for model architecture have been explored. I believe this work represent a substantive advance in modelling science. Below I list specific comments which I hope will serve to improve the manuscript.

Response: We really appreciate the reviewer's recognition of our work and all other valuable comments and suggestions mentioned below. Just as the reviewer summarized, we want to incorporate the domain knowledge learned from agroecosystem process-based model *ecosys* to the advanced machine learning models to combine the advantages from both kinds of state-of-art works. This effort is trying to build a new body of research for simulating the agriculture ecosystem and KGML-ag in this study is a demonstration case simulating $N_2O$ flux from mesocosm experiments. To further improve our study, we have carefully revised the manuscript to address all reviewer's comments. The specific responses can be found in the following letters.

Specific comments

1. The use of the term "initials" confuses me. Upon first reading I thought it referred to the acronyms for various intermediate variables. I think it actually refers to the initial values of a sequence. Is this usage standard? If not, I recommend a different phrase such as "initial values" in place of the word "initials." Alternatively, clarify the meaning of the term in the manuscript.

Response: Thanks so much for pointing out this term which may cause confusion. Just as you said, the term "initials" in the manuscript are most referring to the "initial values". It indeed will cause some confusion since we also use the term "initial" as a verb for the knowledge guided initialization. Thus we have replaced "initials" to "initial values" throughout the manuscript.

2. Another possible explanation for why KGML-ag2 better predicts IMVs but does not predict N2O as well is that KGML-ag1 may learn to use the IMVs as a kind of extra hidden layer, encoding information relevant to N2O predictions in them.

Response: We really appreciate your interesting explanation about why KGML-ag2 predicts better IMVs but worse $N_2O$ fluxes. In both KGML-ag1 and KGML-ag2, the IMVs were first predicted from KGML-ag-IMV modules and then input into the KGML-ag-$N_2O$ modules. The only difference between

KGML-ag1 and KGML-ag2 is that KGML-ag2 explicitly simulates each IMV by using individual KGML-ag-IMV modules. Thus, using IMVs as a kind of extra hidden layer may happen in both models in KGML-ag-$N_2O$ modules. But since KGML-ag1 has interactions between predicted IMVs and lower complexity, it may be easier for the KGML-ag1-$N_2O$ module to get the useful knowledge from IMVs.

Moreover, your valuable thought draws us to deeply review the model structures and data qualities. The observational data, including the IMVs of $CO_2$, $NH_4^+$, $NO_3^-$ and VWC, are not perfect and may have many noises or be lacking some key information. KGML-ag2-IMV module may only follow what we have for IMVs to generate accurate IMV predictions without any extra information, while KGML-ag1-IMV module may perform like an encoding layer to predict IMVs with extra information relevant to $N_2O$ flux, just as you mentioned.

In this revision, we decided to keep our explanation to make our discussion more focused and accessible to a broader audience. But we will find a larger dataset to test both explanations in subsequent ML-oriented technical papers.

3.  Why not include KGML-ag2 in Figure 4? I can see simplifying the comparison by choosing only the best-performing model.

Response: The reviewer is right that we excluded KGML-ag2 in the previous Figure 4 to simplify the comparison. To address the reviewer's concern, we have added similar 10 ensemble experiments for KGML-ag2 and updated Figure 4 (also as Figure R5 below). We can see that although KGML-ag2 has similar mean performance as the KGML-ag1 but it has much larger uncertainties. Moreover, the best scores for slope and curvature are all from KGML-ag1.

[Figure]

Figure R5: The comparisons of overall prediction accuracy for $N_2O$ value (a), 1st order gradient (slope, b) and 2nd order gradient (curvature, c) between four tree-based ML models (DT, RF, GB and XGB), two deep learning models (ANN, GRU), and KGML-ag models. Different color symbols represent the different models. The x- and y-error bars are coming from the maximum and minimum scores of ensemble experiments. The dot represents the mean score of the ensemble experiments.

We have also updated the corresponding figures including Figure 5, Figure S6-S7, and section 3.3 (From line xxx to yyy).

"The results from eight different models showed that KGML-ag1 comparing with other pure ML models consistently provided the lowest RMSE (3.59-3.94 mg N m$^{-2}$ day$^{-1}$, 1.14-1.23 mg N m$^{-2}$ day$^{-2}$, and 0.84-0.89 mg N m$^{-2}$ day$^{-3}$) and highest r$^2$ (0.78-0.81, 0.48-0.56, and 0.23-0.31) for N$_2$O fluxes, slope and curvature, respectively (Fig. 4). This indicated that KGML-ag1 outperformed other pure ML models in capturing both the magnitude and dynamics of N$_2$O flux. KGML-ag2 presented slightly better mean scores for N$_2$O flux predictions than KGML-ag1, but worse scores for slope and curvature and larger uncertainties. This proved the hypothesis discussed in section 3.2 that KGML-ag2 didn't benefit the magnitude and dynamics predictions of N$_2$O flux with its more complex structure and less connections between IMVs"

4. Many standard deep learning models were included for comparison, but an LSTM was not among them. I would expect the LSTM to perform similarly to the GRU. I don't think it is crucial that an LSTM be included in this comparison. However, if the GRU outperforms an LSTM, it could provide further justification for choosing to use a GRU instead of an LSTM. Again, I could understand simplifying the comparison by including only one recurrent neural network.

Response: We fully agree with your comments on LSTM. We have tested both GRU and LSTM as mentioned in section 2.2.3, and preliminary results showed similar performance between the two neural network structures. However, to simplify the comparison and streamline the discussion, we fixed GRU as the basis for pure machine learning models and the KGML models.

To address the reviewer's concern, we have conducted similar 10 ensemble experiments of LSTM and the comparisons are presented here in Figure R1 and in the supplement Figure S8 (best model in ensemble experiment). We can see in Figure R1 that the LSTM is slightly better than GRU in predicting N$_2$O flux value and similar as KGML-ag1. But for slope and curvature predictions, LSTM is similar to GRU and KGML-ag1 can always outperform LSTM. From Figure S8 demonstration case, the LSTM with r$_L^2$ of 0.72 and r$_U^2$ of 0.73 is better than GRU model (r$_L^2$ of 0.60 and r$_U^2$ of 0.57) but worse than KGML-ag1 (r$_L^2$ of 0.78 and r$_U^2$ of 0.86). This further proved our conclusion that KGML-ag1 better represents complex dynamics of N$_2$O flux than other pure machine learning models.

5. You tested two input combinations, IMVcb1 and IMVcb2, but it is not clear how that test informed the model development.

Response: Thank you for finding this unclear part in our manuscript. We have added more descriptions to clarify why we have tested two combinations in section 2.2.4 paragraph 2 (From line xxx to yyy).

"Variables ranked high in feature importance analysis should be primarily considered during model development. To develop a functionable KGML-ag in real world, we further investigated the feature importance of four IMVs that are available from mesocosm observations including CO$_2$, NO$_3^-$, VWC and NH$_4^+$, which were ranked 7th, 20th, 58th, 60th respectively in 92 input features of synthetic data (Fig. S2a). We used these four available IMVs to create two input combinations: 1) CO$_2$ flux, NO$_3^-$, VWC and NH$_4^+$ (IMVcb1), and 2) NO$_3^-$, VWC and NH$_4^+$ (IMVcb2). The objective of building IMVcb2 was to investigate the importance of highly ranked variable CO$_2$ flux (by removing it from the inputs), and the impact of mixing-up flux and non-flux variables on model performance. "

Moreover, tests using IMVcb1 (with $CO_2$) and IMVcb2 (without $CO_2$) indicate that high ranking variables detected from feature importance analysis based on synthetic data (like $CO_2$ flux ranks 7th in 92 input features ) can also be similarly important in $N_2O$ predictions with real observed data. Therefore the feature importance results could benefit feature selection in real data. We have added the results and discussion in section 3.2 last paragraph (From line xxx to yyy).

"In addition, we also found all KGML-ag models would perform better by using IMVcb1 (with $CO_2$) than using IMVcb2 (without $CO_2$) in real data tests, indicating feature importance analysis based on synthetic data can be a reasonable substitute for analysis with the often limited real-world data."

6. The reason for evaluating slope and curvature in addition to N2O value could be stated more clearly.

Response: We have added more explanations in section 3.2 paragraph 2 (From line xxx to yyy).

"Slope represents the speed of $N_2O$ flux changes through time and curvature represents the acceleration. Assessing prediction performance on these two metrics will reveal the model robustness on capture variable dynamics, which is critical when predicting fast-change variables with hot moments like $N_2O$."

7. I recommend that the paragraph starting at line 194 be rewritten for clarity. First, data augmentation is a class of methods, not a single method. Second, Meyer et al. use copula-based models in particular to augment datasets. Do you use copula-based methods? The way this reference is cited suggests that you follow their approach. Third, do you randomly sample observed data, or synthetically generated data, or both? Do you randomly sample only the data which are hourly, e.g., air temperature, net radiation, N2O, CO2, and VWC? How is the daily value calculated from the sampled data? I did not find the answers to these questions to be clear from the text.

Response: We really appreciate your detailed comments on the data augmentation method. In this revision, we have deleted the confusing sentence "Data augmentation is a typical practice in ML when training data is limited (Meyer et al., 2021)" because we did not intend to highlight one particular method, but only to explain the data augmentation concept using one recent citation. To your second question, the augmentation method is only used on observed data and corresponding weather forcings. To your third question, we only randomly sample the data which are hourly. Lastly, we used the average of the 16 hours (or maximum valid hours) of data to represent the daily values. We have addressed all those questions in the new paragraph:

"To reduce overfitting and increase the generalization of the trained model based on the small amount of mesocosm data, we applied the following method to augment the experimental measurements and weather forcings to 1000 times larger by sampling hourly data and averaging them to daily scale. In this method, 16 hours (or maximum valid hours) of data are randomly selected from 24 hours of data to compute their mean as the daily value. Since 3/4 of the day are covered by the selected data (16 hours /24 hours), the augmented daily values should be representative enough for the source day and meanwhile present slight variations. Furthermore, the observation ratio, (24 hours - missing hours) / 24 hours, can be used as the weights in loss function to inject the data quality information in model optimization. If the day

has more than 16 hours missing values, we consider the observations in that day as not trustworthy and drop the day by setting the weight to 0. This method can not only augment the data to 1000 times larger but also deal with the missing values in observed data inherently. The total amount of observed mesocosm data and related weather forcings are augmented to 122 days x 3 years x 6 chambers x 1000 data samples in this study."

8. How well does the model perform out-of-sample? Out-of-sample performance is mentioned in the introduction, but the discussion does not address it.

Response: We totally agree with the reviewer that out-of-sample performance would be critical for predictive models. Thus we have mentioned in the introduction that out-of-scenario ability is the limitation of machine learning models. In our study, we have compared the out-of-sample performance between different models using the period without any observation data in section 3.2 paragraph 1:
"For the region without observation data (normally before day 25), KGML-ag1 predicted stable $N_2O$ fluxes close to 0 mg N $m^{-2}$ $day^{-1}$ (which is close to the reality in the experiment setting) while GRU caused anomalous peaks of fluxes. This is because KGML-ag1 has learned knowledge for the whole period from the pretraining process with *ecosys* model generated synthetic data, but GRU model has no prior knowledge for the period without any data in observations;"
and section 3.3 last paragraph:
"From these comparisons, we infer that without considering temporal dependence and pretraining process, the tree-based model including DT, RF, GB and XGB and deep learning model ANN predicted erratic peaks in almost every missing data point, while GRU model was stable in small gaps and only presented poor performance in long missing period (before 25 day). This improvement by GRU model can be attributed to the structure of GRU that naturally keeps the historical information using hidden states, which enables GRU to consider the temporal dependence and make consistent predictions over time."
Moreover, the objective for this study is to explore ways to incorporate knowledge into ML models for improving agriculture ecosystem simulation. The mesocosm experiment measured many inputs and intermediate variables in addition to the output of $N_2O$ fluxes, thus serving as a unique testbed. Continuous $N_2O$ flux data with a comprehensive set of input and intermediate variables, especially those at hourly or daily scales, are very limited. Some recent projects funded by the US Department of Energy have started to collect such datasets in real-world fields, but the data has not been released. While we fully understand the importance of out-of-sample testing, working with another dataset is beyond the scope of this manuscript.

Technical corrections

● At line 239, Sec. 4.4 does not exist.

Response: We have corrected the sentence by replacing 4.4 to existing 4.3..
"... and would guide future N2O related measurements and KGML model development (discussed in Sec. 4.3)."

● At line 240, I believe this should refer to Fig. 1c and 1d, not 1b and 1c.

Response: We have corrected this mistake.
"Next we used the knowledge learned from synthetic data to develop the structure of KGML-ag (Fig. 1c-d)."

- Tables 1 and 2 have identical captions but different contents.

Response: We have corrected this by replacing the right caption.
"Table 2: Prediction accuracy comparisons between non-pretrained GRU model and KGML-ag1."

- Sections 4.1 and 4.2 are both entitled "Interpretability of KGML-ag."

Response: We have replaced the section 4.2 title to "Lessons for KGML-ag development"

**To Reviewer 3**

The authors are proposing the development of a new approach KGML-ag to machine learning in estimating N2O emissions from fertilized agricultural fields. This approach involves using data generated from a process model and a mesocosm experiment to tune the relationships and their parameters among input and intermediate variables by which N2O emissions are thought to be governed. The advantages of this approach over process models are simplified input data requirements, more rapid model execution, and possibly more accurate simulation of N2O fluxes measured in experiments for which the model is tuned.

Response: We really appreciate the reviewer correctly recognizing our efforts and achievements. We want to incorporate the domain knowledge learned from agroecosystem process-based model *ecosys* to the advanced machine learning models to combine the advantages from both. Developing KGML-ag is one of the very first few attempts to realize the concept of hybrid modeling (Reichstein et al. 2019 Nature) in simulating agroecosystem biogeochemistry. To further improve our manuscript, we have carefully revised the content based on all reviewers' comments and suggestions.

The ability of this approach to simulate N2O emission events under controlled laboratory conditions is impressive. It should be noted that the N2O emissions in Fig. 2 and the soil NO3 contents in Fig. 3 are much larger than those commonly encountered in field conditions. However the relationships and their parameters upon which this approach is based are not disclosed to the reader, and so remain a 'black box'. For example, in section 4.1 the processes governing the time course of N2O emissions following a urea application are described, but the method by which these processes were represented in KGML is not.

Response: We have double checked the $N_2O$ emission and $NO_3^-$ concentration magnitude from mesocosm and comparing with other field studies under similar conditions (Fassbinder et al., 2013; Grant et al., 1999, 2006, 2008; Hamrani et al., 2020; Venterea et al., 2011). It turned out that our magnitude for $N_2O$ (peak value around 20 mg N $m^{-2}$ $day^{-1}$) and $NO_3^-$ (peak value around 50 g N $m^{-2}$) are within the field observed ranges for managed crop soils. The reviewer's impression that these values being "too large" is likely because of the different units we used. Here all units are converted to daily scale as a default setting in *ecosys*, while other studies often report N fluxes using mg N $m^{-2}$ $h^{-1}$ for $N_2O$ flux and mg N $kg^{-1}$ for $NO_3^-$ concentration (in this case, peak values in our experiment are 1 mg N $m^{-2}$ $h^{-1}$ and 40 mg N $kg^{-1}$). To avoid future misunderstandings of the data, we first add a sentence in data description section 2.2.2 to include the comparisons with other studies (From line xxx to line yyy) and then add units in Figure 2 and Figure 3 caption to notify readers about the different units being used.

"The magnitude of $N_2O$ flux and $NO_3^-$ soil concentration and their responses following fertilizer application from this mesocosm experiment are consistent with several field studies of agricultural soils (Fassbinder et al., 2013; Grant et al., 1999, 2006, 2008; Hamrani et al., 2020; Venterea et al., 2011)."
"Figure 2: $N_2O$ flux time series comparisons among pure non-pretrained GRU predictions (blue line), KGML-ag1 predictions (red line) and observations (black line-dot) from cross-validation. The $N_2O$ flux unit is mg N $m^{-2}$ $day^{-1}$."
"Figure 3: IMVs prediction from KGML-ag1. The black-dot line represents observations and the red line represents the results from KGML-ag1. Chmb is the abbreviation for chamber. $r^2$ and RMSE are

calculated and present in each year and chamber. The $CO_2$ flux and soil $NO_3$- concentration units are g C $m^{-2}$ $day^{-1}$ and g N $m^{-2}$, respectively."

"Figure 3 Contd.: IMVs prediction from KGML-ag1. The black-dot line represents observations and the red line represents the results from KGML-ag1. Chmb is the abbreviation for chamber. $r^2$ and RMSE are calculated and present in each year and chamber. The soil $NH_4^+$ concentration and soil VWC units are g N $m^{-2}$ and $m^3$ $m^{-3}$, respectively."

We would like to note that this study is one significant step towards none-black box use of machine learning, but fully opening the black box is one of the frontiers in ML research that still has a long way to go. We partially opened the black box by incorporating domain knowledge into a completely black box ML model via three efforts: 1) building a hierarchical structure (with black-box GRU model as basis) to simulate the important intermediate variables (IMVs) first; then the predicted IMVs are used as the additional inputs in target variable simulation (e.g. $N_2O$), which will provide an opportunity to track those IMVs during the simulation period; 2) pretraining the KGML model with a process-based model so that the KGML model can perform as a surrogate model of the process-based model; 3) other techniques like using initial values to preserve state, feature importance analysis and stepwise training and fine tuning etc. With these implementations, our KGML model not only outperformed pure ML models but also was more interpretable. The ability to predict IMVs also shed light on model improvement, which is not possible or much more complicated with pure ML models.

Regarding the relationships and parameters, we will make the KGML-ag code and neural network weights open through Github once the review process is done. But explicitly describing these like what is often done for process-based models is not practical because KGML-ag is essentially a neural network model, and readers are not able to infer much directly from layers, nodes and weights.

Finally, we agree with the reviewer that in some cases why KGML performed so well needs to be explained, but this would not deny our contribution towards opening the "black box". To reflect the reviewer's concern, we have added in the discussion section 4.3 last paragraph that:

"Finally, at the current stage we can not claim to have completely opened the black box of KGML-ag, but this framework is a significant step towards this goal. For example, some ideas implemented in our study, such as using pretraining to transfer knowledge from PB model to ML model, incorporating causal relations by hierarchical structure, predicting IMVs for tracking middle changes and using initial values as input to reduce data demand, would shed light on the future KGML-ag model improvement."

As for all black box approaches to modelling, it is vitally important that KGML be subjected to tests with truly independent datasets, i.e. datasets that are completely separate, and preferably very different, from those used in model calibration. Impressive results can always be achieved by calibrating enough parameters, but are these parameters robust? The extent to which such testing of KGML was conducted in this paper is not clear. At the very least, for this paper to be publishable, calibration and validation of KGML must be clearly distinguished, and clear evidence of independent testing must be provided. Further description of the key relationships and their parameters that govern N2O emissions in the model should also be provided so as to improve confidence in its robustness.

Response: We agree with the reviewer that out-of-sample testing is critical for model development. In this work all results reported in Figure 4 and Figure 5 are from leave-one-out experiment. For example, we trained KGML with data from chamber 1-5 and tested it against the left out chamber 6 as the model performance. Another out-of-sample test is by comparing the prediction performance during the periods without any chamber observation data (i.e. before April 25th of each year). Results show that KGML-ag1 predicted stable $N_2O$ fluxes close to 0 mg N m$^{-2}$ day$^{-1}$ (which is close to the reality in the experiment setting) while GRU caused anomalous peaks of fluxes. This highlighted the power of KGML because KGML-ag1 has learned "knowledge" for the whole period from the pretraining process using *ecosys* model generated synthetic data. Relevant text can be found in xxx-yyy:

"For the region without observation data (normally before day 25), KGML-ag1 predicted stable N2O fluxes close to 0 mg N m-2 day-1 (which is close to the reality in the experiment setting) while GRU caused anomalous peaks of fluxes. This is because KGML-ag1 has learned knowledge for the whole period from the pretraining process with *ecosys* model generated synthetic data, but GRU model has no prior knowledge for the period without any data in observations;"

and in lines xxx-yyy:

"From these comparisons, we infer that without considering temporal dependence and pretraining process, the tree-based model including DT, RF, GB and XGB and deep learning model ANN predicted erratic peaks in almost every missing data point, while GRU model was stable in small gaps and only presented poor performance in long missing period (before 25 day). This improvement by GRU model can be attributed to the structure of GRU that naturally keeps the historical information using hidden states, which enables GRU to consider the temporal dependence and make consistent predictions over time."

We understand these two out-of-sample tests are not in the sense of being "very different" from what the KGML model was developed. However, this is so far the best data we can access. The mesocosm experiment data we used in this study has provided a comprehensive set of inputs and intermediate variables in addition to the output of N2O fluxes, thus serving as a unique testbed. Continuous $N_2O$ flux measurements along with a comprehensive set of input and intermediate variables, especially those at hourly or daily scales, almost do not exist or are not publicly accessible. Some recent projects funded by the US Department of Energy have started to collect such gold standard dataset under field conditions, but the data needs to be accumulated for another one or two years before release. We anticipate that gold standard data will significantly benefit the development of the KGML-ag model.

Finally, we argue that the novelty and robustness of our study can be justified in a different perspective. Our results show that a well-calibrated *ecosys* is not able to reproduce many dynamics of observed $N_2O$ fluxes (Fig. S9) regardless how we tune *ecosys* parameters. A pure ML model can better reproduce the time series, but still has missed several key peaks in growing season while falsely predicted spring peak emissions even though fertilizers were not applied until several days later (Fig. 2). The KGML-ag1 leveraged the advantage of *ecosys* and the pure ML model, and outperformed both (Fig. 2). These nested comparisons clearly demonstrate the power of KGML as a framework. While we do not argue that KGML-ag is a perfect model that would be directly applicable to other places, sharing our approach will provide food-for-thought to the community on how to build a hybrid biogeochemical model that is computationally more efficient and more robust than both process-based and ML-based models. We have added new discussions about this concern in the last paragraph of section 4.3 (from line xxx to yyy).

"Besides, we acknowledge the importance of further testing the KGML-ag over completely independent datasets, but results presented in this manuscript are sufficient to justify the power of KGML as a framework. The mesocosm experiment data we used in this study has provided a comprehensive set of inputs and intermediate variables in addition to the output of $N_2O$ fluxes, thus serving as a unique testbed. We expect our validation results will be more solid once more gold standard data of $N_2O$ fluxes along with other relevant inputs and intermediate variables become publicly available."

In the Discussion, the authors rightfully address some of the factors that may limit the robustness of KGML. These limitations will likely become more apparent when the authors conduct tests of KGML under field conditions. Addressing these factors, as described by the authors, appears to require that KGML more closely resemble process-based models, and may reduce the computational advantages claimed for the KGML approach.

Response: The reviewer's concern on decreased performance in field application is legit, and is a good hypothesis to test when more dataset become available. At this stage, we do not know whether or not these limitations will become more apparent under field conditions. But we are currently collecting new gold standard data of inputs, intermediate and $N_2O$ fluxes from both field and lab experiments, which will be used to test the reviewer's hypothesis. We would also like to acknowledge that KGML-ag's limitations apply to both pure ML model and process-based models under field conditions, so it is very likely KGML-ag will continue to outperform both.

       Another concern by the reviewer is that further development of KGML will make it resemble process-based models, thereby reducing the computational advantages. We argue this is unlikely because the application of neural networks is faster than process-based models by multiple orders. To surrogate as many components of process-based models as possible is one research frontier in hybrid modeling for earth system science (Reichstein et al. 2019 Nature; Irrgang et al. 2021 Nature Machine Intelligence), with latest advances occurred in weather forecast (Bauer et al. 2021 Nature Computational Science). By using a hybrid model, computationally inefficient components of PB can be identified one by one, and be replaced with more efficient ML-based surrogates to eventually obtain the most efficient model, thereby resolving the concern raised by the reviewer. We have added the new discussion at the end of section 4.3 to address the reviewer's concern (from line xxx to yyy).

"Moreover, incorporating more and more domain knowledge into KGML-ag will be inevitable in further improvement, but we don't think KGML-ag will become inefficient as it becomes more like the PB model. In fact, to efficiently surrogate components of PB models has been proposed as a research frontier in hybrid modeling for earth system science (Reichstein et al., 2019; Irrgang et al., 2021), with latest advances occurring in weather forecasts (Bauer et al., 2021). By using a hybrid model, computationally inefficient components of PB can be identified one by one, and be replaced with more efficient ML-based surrogates to eventually obtain the most efficient model. Further KGML-ag model development will also need to balance efficiency, accuracy and interpretability."

Reference:

Bauer, P., Dueben, P. D., Hoefler, T., Quintino, T., Schulthess, T. C., & Wedi, N. P.: The digital revolution of Earth-system science. Nature Computational Science, 1(2), 104-113, 2021.

Fassbinder, J. J, Schultz, N. M, Baker, J. M, & Griffis, T. J.: Automated, Low-Power Chamber System for Measuring Nitrous Oxide Emissions, Journal of environmental quality, 42, 606. doi: 10.2134/jeq2012.0283, 2013.

Grant, R. F., & Pattey, E.: Mathematical modeling of nitrous oxide emissions from an agricultural field during spring thaw. Global Biogeochemical Cycles, 13(2), 679-694, 1999.

Grant, R. F., & Pattey, E.: Temperature sensitivity of $N_2O$ emissions from fertilized agricultural soils: Mathematical modeling in *ecosys*. Global biogeochemical cycles, 22(4), 2008.

Grant, R. F., Pattey, E., Goddard, T. W., Kryzanowski, L. M., & Puurveen, H.: Modeling the effects of fertilizer application rate on nitrous oxide emissions, Soil Science Society of America Journal, 70(1), 235-248, 2006.

Hamrani, A., Akbarzadeh, A., & Madramootoo, C. A.: Machine learning for predicting greenhouse gas emissions from agricultural soils, Science of The Total Environment, 741, 140338, 2020.

Irrgang, C., Boers, N., Sonnewald, M., Barnes, E. A., Kadow, C., Staneva, J., & Saynisch-Wagner, J.: Towards neural Earth system modelling by integrating artificial intelligence in Earth system science. Nature Machine Intelligence, 3(8), 667-674, 2021.

Reichstein, M., Camps-Valls, G., Stevens, B., Jung, M., Denzler, J., & Carvalhais, N.: Deep learning and process understanding for data-driven Earth system science. Nature, 566(7743), 195-204, 2019.

Venterea, R. T., Maharjan, B., & Dolan, M. S.: Fertilizer source and tillage effects on yield‑scaled nitrous oxide emissions in a corn cropping system. Journal of Environmental Quality, 40(5), 1521-1531, 2011.

---

## Editor Decision (ED1)

The authors have done a good job responding to reviewer comments and concerns. There are some remaining improvements to be made, I think, but these are relatively minor. Thus, I recommend **publication with minor revisions** (with a final review by me).

**Primary comments**

**Reviewer 1 comment 2**

Reviewer 1 asked specifically about *ecosys* performance with regard to high-frequency fluctuations. It's unclear whether that's included in the $R^2$ value for NEE listed on line 151—what frequency was that analysis at?

There's nothing in the new text about $N_2O$ performance; please add some text pointing to Grant et al. papers as you did for in the response to the reviewer (although that level of detail isn't necessary): "1) the papers of Grant et al (2006, 2008) to find the influences of fertilizer rate and temperature on $N_2O$ emissions in fertilized agriculture soil; 2) the paper of Grant et al (1999) to find the influences of spring thawing; and 3) the papers of Grant et al (2010, 2016) to check the $N_2O$ simulation performances at managed forest and grassland."

The Wang et al. (2021) reference is missing from the References, and I can't check Yang et al. (2022) because it's only been submitted.

**Reviewer 1 comment 3**

Yes, you cited the Miller (2021) thesis, which has chamber measurement uncertainty. But I think the reviewer was saying it would be good to explicitly compare the uncertainty in the simulations to the uncertainty in the chamber observations. Please add some discussion of this.

**Reviewers 2 and 3: Out-of-sample performance**

The reviewers seem to have missed that Chamber 6 served as an out-of-sample evaluation. This should be made clearer throughout the manuscript.

- Figs. 2 and 3 should indicate (graphically and in caption) which chamber was out-of-sample
- Figs. 4 and 5 should *only* include the out-of-sample chamber as the observation. (It's unclear whether this is already the case.) This should be mentioned in their captions.
- Same for Tables 1–3.

You also point to how the model behaves when there is no chamber observation data as an additional out-of-sample test. While you may *suspect* it does well, without any measurement data, it's not justifiable to use this as a certain measure of performance. Please revise lines 424–429 to reflect that (e.g., "poor **assumed** performance," "**assumed** improvement).

**Reviewer 3: $N_2O$ fluxes and $NO_3$ concentrations higher than normal**

In your reply to Reviewer 3, you posited that you saw peaks of $N_2O$ around 20 mgN $m^{-2}$ $day^{-1}$ and $NO_3$ around 40 mgN $kg^{-1}$. However, it appears from Figs. 2 and 3 that those are actually about 60 mgN $m^{-2}$ $day^{-1}$ and 95 gN $m^{-2}$, respectively. Compare to the papers you cited (units converted as necessary to match yours):

| $N_2O$ emissions (mgN $m^{-2}$ $day^{-1}$) | Reference |
|---|---|
| 4.8 | Venterea et al. (2011) Figs 3–4 |
| 8.2 | Fassbinder et al. (2013), p. 612 |
| 19.2 | Grant & Pattey (1999) abstract |
| 18 | Grant et al. (2006) Fig. 2 |
| 34 | Grant & Pattey (2008) Fig. 3 |
| 38 | Grant & Pattey (2008) Fig. 4 |
| 50 | Hamrani et al. (2020) |

| $NO_3$ concentration (mgN $kg^{-1}$) | Reference |
|---|---|
| 7.1 | Grant & Pattey (1999) Table 3 |
| 80 | Venterea et al. (2011) Fig. 8 |

It's not a problem that your peaks are higher than seen in other studies, but as Reviewer 3 suggested, this should be disclosed (perhaps in the Discussion).

**Minor corrections**
- L140-1: Should be "respiration, and $NO_2^-$ become**s** **an** alternative electron acceptor" (note not "respirations")
- L143: Should be "considers"
- L216-8:
  - Should be "Since **up to**"
  - 16/24 is ⅔, not ¾
  - Should be "of the day **is**"
- "and meanwhile present slight variations"—it's unclear what this means
- L257: "**the** highly ranked"
- L375: What is a "hot moment"? Please define for less-technical readers.
- L560: "from **a** PB model to **an** ML model"
- L565: "We expect our validation results will be more solid" is a little too casual and vague. Maybe something like, "We expect to further validate and refine our model"
- L567: "Will be inevitable" why?
- L568: "surrogate" isn't a verb. Maybe replace "to efficiently surrogate" with "efficiently emulating"
- Line 723: "structuress"

---

## Author Response (AR2)

**Response Letter to Editor**

We really appreciate all comments and suggestions from the editor and have carefully addressed all concerns and comments point by point. Main changes include:

1) Added more descriptions on *ecosys* model evaluation on $N_2O$ flux;
2) Added discussion on comparisons between mesocosm data uncertainty with model simulation uncertainty;
3) Added more descriptions of leave-one-out cross validation repeatedly to remind reader we only presented validation results;
4) Added discussion on mesocosm data slightly higher than field studies;
5) Removed coloured cell/values as recommended from file validation and corrected other minor mistakes.

We believe the quality of this manuscript has improved after the revision. Below, please find our detailed responses point-by-point.

Please be aware of the formatting of all responses:
1. Reviewer comment in **black**, response in **blue** and quotation from the main text in **red**;
2. The line number is based on the clean version of the revised manuscript, not the track change version

**Detailed responses begin:**

**Post-review decision: gmd-2021-317**

The authors have done a good job responding to reviewer comments and concerns. There are some remaining improvements to be made, I think, but these are relatively minor. Thus, I recommend publication with minor revisions (with a final review by me).

Response:
We are grateful for the editor's recommendation and recognition of our efforts. We have carefully revised the manuscript to address all his comments and suggestions.

**Primary comments:**

**Reviewer 1 comment 2**

Reviewer 1 asked specifically about ecosys performance with regard to high-frequency fluctuations. It's unclear whether that's included in the $R^2$ value for NEE listed on line 151— what frequency was that analysis at?

Response:
Thanks so much for pointing these out. To address this concern, we have revised the related test and added in the frequency information for flux tower NEE and Reco data used in *ecosys* validation (line 148 to 152).

"For the agricultural ecosystems in the US Midwest, whose simulations are used for synthetic data in this study, the performance of *ecosys* on $CO_2$ have been extensively benchmarked, including $CO_2$ exchange (daily Reco $R^2$ = 0.80-0.86; daily NEE, $R^2$ = 0.75-0.89) and leaf area index (LAI, $R^2$ = 0.78) from six flux towers, USDA census reported corn yield ($R^2$ = 0.83) and soybean yield ($R^2$ = 0.80), satellite-derived GPP for corn ($R^2$ = 0.83) and soybean ($R^2$ = 0.85) in the US Midwest (Zhou et al., 2021)."

There's nothing in the new text about $N_2O$ performance; please add some text pointing to Grant et al. papers as you did for in the response to the reviewer (although that level of detail isn't necessary): "1) the papers of Grant et al (2006, 2008) to find the influences of fertilizer rate and temperature on $N_2O$ emissions in fertilized agriculture soil; 2) the paper of Grant et al (1999) to find the influences of spring thawing; and 3) the papers of Grant et al (2010, 2016) to check the $N_2O$ simulation performances at managed forest and grassland."

Response: We really appreciate the comments and suggestions from the editor. A new sentence has been added to describe the detailed validation scores for *ecosys* $N_2O$ flux simulation in hourly frequency and to include those Grant et al. papers listed for validation in various ecosystems (line 152 to 155).

"In addition, *ecosys* model can capture the dynamics and magnitude of $N_2O$ flux in hourly frequency ($R^2$ = 0.2-0.4 and RMSE = 0.1-0.2 mg N m$^{-2}$ h$^{-1}$ in Grant et al., 2008; $R^2$ = 0.28-0.37 and RMSE = 0.2-0.28 mg N m$^{-2}$ h$^{-1}$ in Grant et al., 2003), and in various ecosystems (e.g. agriculture soil in Grant et al., 2006, 2008; forest in Grant et al., 2010; and grassland in Grant et al., 2016)."

The Wang et al. (2021) reference is missing from the References, and I can't check Yang et al. (2022) because it's only been submitted.

Response:

Thanks so much for pointing out these. The Yang et al. (2022) paper and related accumulated $N_2O$ flux validation results have been removed from text since the formal citation is not available at this stage. Besides, the actual first author for "Wang et al. (2021)" is Zhou, Wang. Thus we have corrected the citation to "Zhou et al. (2021)" and added the reference to the list (line 732 to 734).

"Zhou, W., Guan, K., Peng, B., Tang, J., Jin, Z., Jiang, C., ... & Mezbahuddin, S.: Quantifying carbon budget, crop yields and their responses to environmental variability using the ecosys model for US Midwestern agroecosystems. Agricultural and Forest Meteorology, 307, 108521, 2021."

**Reviewer 1 comment 3**

Yes, you cited the Miller (2021) thesis, which has chamber measurement uncertainty. But I think the reviewer was saying it would be good to explicitly compare the uncertainty in the simulations to the uncertainty in the chamber observations. Please add some discussion of This.

Response:

Thanks much for your suggestions. Beside previous added ensemble experiments (presented in section 3.3 and figure 4 and 5), we have added the comparisons of uncertainty distribution ranges and standard deviations between mesocosm observation and KGML-ag1 model simulations. The details can be found in results section 3.3 first paragraph (line 405 to 409):

"Meanwhile, we have calculated the uncertainty of mesocosm measurement due to converting hourly data to daily data during 30-80 days by using augmented value minus mean of the augmented values (-10.2 to 10.4 mg N m$^{-2}$ day$^{-1}$, and standard deviation =1.4 mg N m$^{-2}$ day$^{-1}$). KGML-ag1 during the same period has comparable uncertainties based on ensemble simulations (calculated by ensemble value minus mean of ensemble values; -14.4 to 15.2 mg N m$^{-2}$ day$^{-1}$, with standard deviation = 1.3 mg N m$^{-2}$ day$^{-1}$)."

**Reviewers 2 and 3: Out-of-sample performance**

The reviewers seem to have missed that Chamber 6 served as an out-of-sample evaluation. This should be made clearer throughout the manuscript.

• Figs. 2 and 3 should indicate (graphically and in caption) which chamber was out-of-sample

• Figs. 4 and 5 should only include the out-of-sample chamber as the observation. (It's unclear whether this is already the case.) This should be mentioned in their captions.

• Same for Tables 1–3.

Response:

We really appreciate your suggestions. Indeed we should repeatedly remind reviewers and readers that model performance is evaluated based on a leave-one-out cross validation (LOOCV) method. Each time when we trained and evaluated a model, the six chambers' data were split into five chambers for training and the remaining one as validation, thus serving as out-of-sample evaluation. Following the editor's suggestion, we have first added a sentence in method section 2.2.2 paragraph 2 to explain the validation process (line 210 to 213):

"We used the leave-one-out cross-validation (LOOCV) method for the evaluation process. Each time we used five chambers' data for model finetuning and another one chamber data for validation. For example, if we used chamber 1-5 to train the model, then chamber 6 would serve as the out-of-sample data to validate the results. Only the validation results would be presented in our study."

Moreover, we have added descriptions about LOOCV being used and all results are from validation data sets to 2-5 figure captions:

"Figure 2: Leave-one-out cross validation of time series of $N_2O$ flux (mg N $m^{-2}$ $day^-$) predicted by the pure non-pretrained GRU model (blue line) and KGML-ag1 model (red line). Observations are shown as black line-dots. Validation results for each chamber were based on out-of-sample predictions by models trained by other five chambers."

"Figure 3: Leave-one-out cross validation of time series of IMVs predicted by KGML-ag1 model (red line). Observations are shown as black line-dots. Validation results for each chamber were based on out-of-sample predictions by models trained by other five chambers. Chmb is the abbreviation for chamber. $r^2$ and RMSE are calculated and present in each year and chamber. The $CO_2$ flux and soil $NO_3^-$ concentration units are g C $m^{-2}$ $day^{-1}$ and g N $Mg^{-1}$, respectively. "

"Figure 3 Contd.: Leave-one-out cross validation of time series of IMVs predicted by KGML-ag1 model (red line). Observations are shown as black line-dots. Validation results for each chamber were based on out-of-sample predictions by models trained by other five chambers.Chmb is the abbreviation for chamber. $r^2$ and RMSE are calculated and present in each year and chamber. The soil $NH_4^+$ concentration and soil VWC units are g N $Mg^{-1}$ and $m^3$ $m^{-3}$, respectively."

"Figure 4: The comparisons of overall prediction accuracy from leave-one-out cross validation for $N_2O$ value (a), 1st order gradient (slope, b) and 2nd order gradient (curvature, c) between four tree-based ML models (DT, RF, GB and XGB), two deep learning models (ANN and GRU) and KGML-ag models. The overall performances were calculated by comparing out-of-sample predictions (each chamber's predictions were from models trained by other five chambers) from all validated chambers with observations. Different color symbols represent the different models. The x- and y-error bars are coming from the maximum and minimum scores of ensemble experiments. The dot represents the mean score of the ensemble experiments. "

"Figure 5: The comparisons of $N_2O$ flux prediction accuracy $r^2$ (a) and (b) RMSE from leave-one-out cross validation, between four tree-based ML models (DT, RF, GB and XGB), two deep learning models

(ANN and GRU) and KGML-ag models in six chambers. Validation results for each chamber were based on out-of-sample predictions by models trained by other five chambers. The gray error bars are coming from the maximum and minimum scores of ensemble experiments."

and table 2 footnote a:
"[a]Leave-one-out cross validation results for each chamber were based on out-of-sample predictions by models trained by other five chambers. The "All data" performances were calculated by comparing out-of-sample predictions from all validated chambers with observations. "
and table 3 footnote c:
"[c]The leave-one-out cross validation overall performances were calculated by comparing out-of-sample predictions (each chamber's predictions were from models trained by other five chambers) from all validated chambers with observations."
Table 1 presented the pretrained results from synthetic data and it used ecosys model generated data to train and evaluate the model. Only the results from testing data (synthetic testing data from 19 counties) were presented. We have also added this information into the table 1 title:
"Table 1: Pretrain results for different model and IMV combinations using *ecosys* synthetic data. Only performances from testing data sets (synthetic data from 19 counties) were presented. "

You also point to how the model behaves when there is no chamber observation data as an additional out-of-sample test. While you may suspect it does well, without any measurement data, it's not justifiable to use this as a certain measure of performance. Please revise lines 424–429 to reflect that (e.g., "poor assumed performance," "assumed improvement).
Response:
Thanks so much for the suggestion. We have added a new sentence at the beginning to declare this assumption and revised the part in section 3.3 last paragraph to reflect the *assumed* performance increase (line 431 to 438):
"For periods without any observed data, we assumed that the good model predictions should be stable, consistent with the nearest period and close to the reality in the experiment setting (e.g. no erratic peak and $N_2O$ flux near 0 mg N $m^{-2}$ $day^{-1}$ before day 25). From these comparisons, we infer that without considering temporal dependence and pretraining process, the tree-based model including DT, RF, GB and XGB and deep learning model ANN predicted erratic peaks in almost every missing data point, while the GRU model was stable in short missing period (1-2 days of missing data) and only presented poor performance in long missing period (before day 25). This improvement by the GRU model may be attributed to the structure of GRU that naturally keeps the historical information using hidden states, which enables GRU to consider the temporal dependence and make consistent predictions over time."

**Reviewer 3: $N_2O$ fluxes and $NO_3$ concentrations higher than normal**
In your reply to Reviewer 3, you posited that you saw peaks of $N_2O$ around 20 mgN $m^{-2}$ $day^{-1}$ and $NO_3$ around 40 mgN $kg^{-1}$. However, it appears from Figs. 2 and 3 that those are actually about 60 mgN $m^{-2}$ $day^{-1}$ and 95 gN $m^{-2}$, respectively. Compare to the papers you cited (units converted as necessary to match yours):

| $N_2O$ emissions (mgN m$^{-2}$ day$^{-1}$) | Reference |
|---|---|
| 4.8 | Venterea et al. (2011) Figs 3–4 |
| 8.2 | Fassbinder et al. (2013), p. 612 |
| 19.2 | Grant & Pattey (1999) abstract |
| 18 | Grant et al. (2006) Fig. 2 |
| 34 | Grant & Pattey (2008) Fig. 3 |
| 38 | Grant & Pattey (2008) Fig. 4 |
| 50 | Hamrani et al. (2020) |

| $NO_3$ concentration (mgN kg$^{-1}$) | Reference |
|---|---|
| 7.1 | Grant & Pattey (1999) Table 3 |
| 80 | Venterea et al. (2011) Fig. 8 |

It's not a problem that your peaks are higher than seen in other studies, but as Reviewer 3 suggested, this should be disclosed (perhaps in the Discussion).

Response:

We really appreciate the editor's efforts on checking the details of the $N_2O$ flux and $NO_3^-$ soil concentration in the cited references. We have also double-checked all numbers in the references and our mesocosm data set. The $N_2O$ fluxes from all six mesocosm chambers during the peak regions (45-60 days) are 16.9±11.7 mg N m$^{-2}$ day$^{-1}$, except for abnormal values (> 40 and up to 71 mg N m$^{-2}$ day$^{-1}$) in 2016 of chamber 3 and 4. The $NO_3^-$ soil concentrations from all six mesocosm chambers during the peak regions (45-60 days) are 59.3±20.7 g N Mg$^{-1}$, with highest values up to 95.2 mg N m$^{-2}$ day$^{-1}$. $NO_3^-$ data from our mesocosm experiments are from 0-15cm but values in Grant et al. (1999) Table 3 are from all layers ranging from 1cm to 115cm. Grant et al. (1999) indicated that the surface layer of the soil would normally have a higher $NO_3^-$ concentration level (31.36 g N Mg$^{-1}$ for 0-15cm). We admit that field level data as listed in references can be different from our mesocosm data (e.g. drainage condition, wind), and our data are in the high end of the ranges summarized from previous field studies.

Following the editor's suggestion, we have revised the description of the data in method section 2.2.2 paragraph 1 and added another field $N_2O$ study (Grant et al., 2016, Fig.4, $N_2O$ peak around 100 mg N m$^{-2}$ day$^{-1}$) to the reference list (line 199 to 202):

"The magnitude of $N_2O$ flux and $NO_3^-$ soil concentration and their responses following fertilizer application from this mesocosm experiment are slightly higher than several field studies of agricultural soils (Fassbinder et al., 2013; Grant et al., 1999, 2006, 2008, 2016; Hamrani et al., 2020; Venterea et al., 2011)."

Besides, we have added discussion in the discussion section 4.3 first paragraph (line 523 to 527):

"The mesocosm measurements of $N_2O$ fluxes (16.9±11.7 mg N m$^{-2}$ day$^{-1}$ during days of 45-60; Highest value is 71 mg N m$^{-2}$ day$^{-1}$) and $NO_3^-$ soil concentrations (59.3±20.7 g N Mg$^{-1}$ during days of 45-60; Highest value is 95.2 g N Mg$^{-1}$) are at the high end of the range that has been observed by field studies (Fassbinder et al., 2013; Grant et al., 1999, 2006, 2008, 2016; Hamrani et al., 2020; Venterea et al., 2011)."

Moreover, we have found that the unit of $NO_3^-$ and $NH_4^+$ soil concentrations (both should be g Mg$^{-1}$ throughout the manuscript) in some places of the main text is not consistent. This is a writing mistake but will not affect any results interpretation. We have fixed them in Figure 3, Figure S1 and Figure S5.

**Minor corrections**

• L140-1: Should be "respiration, and $NO_2^-$ becomes an alternative electron acceptor" (note not "respirations")

Response: We have corrected this in line 139 to 140:

"... when $O_2$ availability fails to meet $O_2$ demand for their respiration, and $NO_2^-$ become alternative electron acceptors."

• L143: Should be "considers"

Response: We have corrected this in line 142 to 143:

"Unlike the pipeline model described by Davidson et al. (2000) , which mainly considers the correlations of $N_2O$ production with nitrogen availability and of $N_2O$ emissions with soil water content, ..."

• L216-8:

      o Should be "Since up to"

      o 16/24 is ⅔, not ¾

      o Should be "of the day is"

      o "and meanwhile present slight variations"—it's unclear what this means

Response: We have corrected the mistakes and revised the sentence to make it more clear in line 218 to 220:

"Since up to 2/3 of the day is covered by the selected data (16 hours /24 hours), the augmented daily values should be representative enough for the source day and with slight variations from each other. "

• L257: "the highly ranked"

Response: We have corrected this in line 258 to 260:

"The objective of building IMVcb2 was to investigate the importance of the highly ranked variable $CO_2$ flux (by removing it from the inputs), and the impact of mixing-up flux and non-flux variables on model performance. "

• L375: What is a "hot moment"? Please define for less-technical readers.

Response: We have added the definition of "hot moment" in line 376 to 378:

"... which is critical when predicting fast-change variables with hot moments (a short period of time with rare events like flux increasing quickly) like $N_2O$."

• L560: "from a PB model to an ML model"

Response: We have corrected this in line 572 to 573:

"... such as using pretraining to transfer knowledge from a PB model to a ML model ..."

• L565: "We expect our validation results will be more solid" is a little too casual and vague. Maybe something like, "We expect to further validate and refine our model"

Response: We have revised the sentence based on the editor's advice in line 578 to 580:

"We expect to further validate and refine our KGML-ag model once more gold standard data of $N_2O$ fluxes along with other relevant inputs and intermediate variables become publicly available."

• L567: "Will be inevitable" why?

Response: We agree with the editor that here "inevitable" is too strong and ignore other possibilities. Thus we have replaced it with "possible" and revised the sentence in line 580 to 581:

"Moreover, incorporating more and more domain knowledge into KGML-ag will be possible for further improvement, ..."

• L568: "surrogate" isn't a verb. Maybe replace "to efficiently surrogate" with "efficiently emulating"

Response: We agree with the editor about this replacement and have revised the sentence in line 581 to 582:

"In fact, to efficiently emulate components of PB models has been proposed as a research frontier in hybrid modeling for earth system science ..."

• Line 723: "structuress"

Response: We have corrected this in figure 1 caption:

"Figure 1: The model structures. ..."

---

## Author Response (AR3)

**Response Letter to Editor**

Please be aware of the formatting of all responses:
1. Reviewer comment in **black**, response in **blue** and quotation from the main text in **red**;
2. The line number is based on the clean version of the revised manuscript, not the track change version

**Comments to the author:**
Looks great, thanks for making those improvements. The only remaining thing is that I'm not sure what you mean with this:

"Meanwhile, we have calculated the uncertainty of mesocosm measurement due to converting hourly data to daily data during 30-80 days by using augmented value minus mean of the augmented values (-10.2 to 10.4 mg N $m^{-2}$ $day^{-1}$, and standard deviation =1.4 mg N $m^{-2}$ $day^{-1}$). KGML-ag1 during the same period has comparable uncertainties based on ensemble simulations (calculated by ensemble value minus mean of ensemble values; -14.4 to 15.2 mg N $m^{-2}$ $day^{-1}$, with standard deviation = 1.3 mg N $m^{-2}$ $day^{-1}$)."

Specifically, the ranges you provide of -10.2 to 10.4 and -14.4 to 15.2: Are those the lower and upper limits of all the differences? I think it would be less confusing to just provide the standard deviations.

Response: We really appreciate the editor's quick response and valuable comments. The editor is correct that the ranges of -10.2 to 10.4 mg N $m^{-2}$ $day^{-1}$ and -14.4 to 15.2 mg N $m^{-2}$ $day^{-1}$ are lower and upper limits of all differences for augmented mesocosm observed data and model simulated data, respectively. It would be good to have those limits to indicate that not only the standard deviations of the uncertainties from observed data and simulated data are similar, but also the ranges are largely overlapped. We totally agree with the editor that it might be confusing if we only include the range values but not any other descriptions. Thus we have added more descriptions to explain the range values and revised the sentences in the section 3.3 first paragraph (line 405 to 410) to clearly express the uncertainty comparisons:
"Meanwhile, we have calculated the uncertainty of mesocosm measurement due to converting hourly data to daily data during 30-80 days by using augmented values minus the mean of the augmented values with lower and upper limits being -10.2 and 10.4 mg N $m^{-2}$ $day^{-1}$, respectively (standard deviation =1.4 mg N $m^{-2}$ $day^{-1}$). KGML-ag1 during the same period has comparable uncertainties based on ensemble simulations with lower and upper limits being -14.4 and 15.2 mg N $m^{-2}$ $day^{-1}$, respectively (calculated by ensemble values minus the mean of ensemble values; standard deviation = 1.3 mg N $m^{-2}$ $day^{-1}$)."